# Sterols and Sphingolipids as New Players in Cell Wall Building and Apical Growth of *Nicotiana tabacum* L. Pollen Tubes

**DOI:** 10.3390/plants12010008

**Published:** 2022-12-20

**Authors:** Nadia Stroppa, Elisabetta Onelli, Patrick Moreau, Lilly Maneta-Peyret, Valeria Berno, Eugenia Cammarota, Roberto Ambrosini, Marco Caccianiga, Monica Scali, Alessandra Moscatelli

**Affiliations:** 1Dipartimento di Bioscienze, Università degli Studi di Milano, Via Celoria 26, 20133 Milan, Italy; 2CNRS, Laboratoire de Biogenèse Membranaire, University of Bordeaux, UMR 5200, 71 Avenue Edouard Bourlaux, 33140 Villenave d’Ornon, France; 3ALEMBIC Advanced Light and Electron Microscopy BioImaging Center, San Raffaele Scientific Institute, DIBIT 1, Via Olgettina 58, 20132 Milan, Italy; 4Dipartimento di Scienze e Politiche Ambientali, Università degli Studi di Milano, Via Celoria 26, 20133 Milan, Italy; 5Dipartimento di Scienze della Vita, Università degli Studi di Siena, Via Aldo Moro 2, 53100 Siena, Italy

**Keywords:** pollen tube, apical growth, *Nicotiana tabacum*, lipid nanodomains, cell wall, vesicle trafficking, actin filaments, clear zone

## Abstract

Pollen tubes are tip-growing cells that create safe routes to convey sperm cells to the embryo sac for double fertilization. Recent studies have purified and biochemically characterized detergent-insoluble membranes from tobacco pollen tubes. These microdomains, called lipid rafts, are rich in sterols and sphingolipids and are involved in cell polarization in organisms evolutionarily distant, such as fungi and mammals. The presence of actin in tobacco pollen tube detergent-insoluble membranes and the preferential distribution of these domains on the apical plasma membrane encouraged us to formulate the intriguing hypothesis that sterols and sphingolipids could be a “trait d’union” between actin dynamics and polarized secretion at the tip. To unravel the role of sterols and sphingolipids in tobacco pollen tube growth, we used squalestatin and myriocin, inhibitors of sterol and sphingolipid biosynthesis, respectively, to determine whether lipid modifications affect actin fringe morphology and dynamics, leading to changes in clear zone organization and cell wall deposition, thus suggesting a role played by these lipids in successful fertilization.

## 1. Introduction

Pollen tube tip growth relies on the coordination of many cell processes involving ion fluxes, cytoskeletal dynamics, membrane trafficking, cell wall formation and turgor pressure [1,2,3,4,5,6,7,8]. The cytoskeleton is a major player in apical growth. In particular, actin filaments (AFs) are responsible for reverse fountain streaming and for delivering/tethering secretory vesicles to specific apical areas, allowing the intake of new cell wall/plasma membrane (PM) constituents and the remodeling of PM/cell wall composition [9,10,11]. In fact, different lipids and proteins are distributed asymmetrically in the apical PM, ensuring correct secretory/endocytic events and vesicle trafficking and, hence, pollen tube elongation [12]. However, the structural constraints regulating polarized secretion and asymmetric protein/lipid distribution on the PM are mostly unknown.

The asymmetric distribution of lipids on the PM has been demonstrated to promote cell polarity in animal, fungal and plant cells [13,14,15,16,17]. Specifically, in somatic cells of *Arabidopsis thaliana*, structural sterols are abundant in the PM and are involved in vesicular recycling and endocytosis, thus emerging as important regulators of root hair tip growth [18]. The requirements for the polarized distribution of sterols and membrane microdomains were recently investigated in Angiosperm pollen tubes: in *Nicotiana tabacum*, sterol-rich domains on the PM, revealed by filipin, are concentrated in the whole apex and in vesicles accumulating in the clear zone, suggesting that sterol-rich membrane domains could be involved in the polarized secretion and in the asymmetric distribution of proteins and lipids [16]. In agreement, use of the styryl dye di-4-ANEPPDHQ made it possible to detect a relationship between low- and high-ordered lipid domains in membranes, highlighting a significantly higher-ordered membrane in the apical region [16,19,20].

These high-ordered lipid microdomains on membranes, called lipid rafts, are well established in animal and plant cells, but their precise functional roles are still subject to debate [21,22,23,24,25,26,27]. Lipid rafts can be characterized in plant cells through isolation of detergent-insoluble membranes (DIMs) [28,29,30]. Purified DIMs from tobacco pollen tube microsomes have a high content of sterols, sphingolipids with very long chain hydroxylated fatty acids, and phospholipids enriched with saturated fatty acids; somatic cell lipid rafts are also rich in polyphosphoinositides, especially Phosphatidylinositol 4,5-bisphosphate (PtdIns(4,5)P_2_) [16,28,29,31,32,33]. Protein analysis of pollen tube lipid DIMs detected proteins that were not identified in microdomains isolated from somatic cell PM: actin, prohibitins and proteins involved in methylation reactions and in phosphoinositide pattern regulation are specifically present in pollen tube DIMs [16]. Pollen tube DIMs also showed proteins that have been identified in microdomains isolated from plant somatic cell PM, such as tubulins, voltage-dependent anion channels and proteins involved in membrane trafficking [16,28,34,35].

The association of actin and tubulin with lipid rafts in plants is noteworthy. Lipid rafts are closely associated with the cortical cytoskeleton and studies using a combination of live-cell imaging techniques have reported that the cortical cytoskeleton influences the formation and/or stability of PM microdomains in plants [36,37,38,39,40]. In particular, while actin filaments are necessary for the dynamic movement of proteins between different membrane phases, microtubules may be responsible for the control of microdomain size and density [38]. Even more interesting, the cell wall/PM/cortical cytoskeleton exists as a cell surface continuum, which is the cell interface with the external world [36]. The cytoskeleton is known to be involved in the maintenance of the cell surface continuum and membrane remodeling, and in turn, a strong association between the cell wall and PM proteins has been reported [41], suggesting interdependence among the functions of the cell wall, cytoskeleton and lipid PM microdomains.

The pollen tube wall consists of a primary pecto-cellulosic wall and a secondary callosic wall, deposited as an inner sheath behind the tip [42,43,44,45]. Pectins appear to be major components of the cell wall at the tube tip. They are secreted at the tip in their highly esterified form [46]. De-esterification of pectins by pectin methylesterase (PME) occurs behind the growing tip; de-esterification exposes pectin carboxyl residues, which can be cross-linked by calcium, increasing cell wall stiffness [47,48], thus regulating dynamic changes in cell wall rigidity/extensibility during pollen tube growth [49,50,51,52]. The cell wall is also stabilized by deposition of cellulose and callose, which together with pectin de-esterification, enhance the stiffness and stability of the cell wall in the shank. Pectin de-esterification with cellulose and callose deposition prevent expansion of the cell wall into the shank and distal region in response to turgor pressure, thus maintaining the cylindrical shape of pollen tubes [53]. Cellulose is normally present in smaller quantities than callose and has also been observed in the apex of pollen tubes, where it probably plays a role in maintaining hemispherical shape [54,55]. Recent evidence suggests that the lipid environment of cellulose synthase complexes (CSCs) is crucial for the proper structural organization and function of the complex itself at the PM. Significantly, CSCs activity has also been identified in DIMs of hybrid aspen cells [56]. However, CesA proteins were not identified in other proteomic studies of DIMs [34,35], presumably due to the large size of CSCs, which makes them difficult to isolate and identify [57]. Although this point remains controversial, it has been postulated that CSCs are delivered to lipid raft microdomains. Callose is synthesized using callose synthase (CalS) [58,59], which is abundant in the pollen tube PM except at the apex, suggesting that callose helps strengthen the cell wall in non-growing regions. Proteomic analysis of DIMs isolated from poplar, tobacco and *Medicago truncatula* reveal that CalS is integrated into the PM, preferentially localizing to detergent-resistant membrane microdomains [35,59,60].

Cell wall precursors, PM components, enzymes and signaling molecules are delivered by highly regulated membrane trafficking and exocytosis. The classical model of tip growth proposes that exocytosis at the pollen tube apex occurs at a higher rate than is required for PM extension [61,62,63]. This means that the maintenance of pollen tube tip architecture requires internalization of excess secreted membrane by endocytosis, while simultaneously time ensuring the correct polarized distribution of proteins and lipids in the PM as the pollen tube elongates. Pollen tube growth can be considered the result of a delicate equilibrium between exocytosis and endocytosis [12,64,65]. The polarized distribution of sterols and liquid-ordered domains in apical-subapical regions of pollen tubes suggests that lipid rafts could be involved in maintaining the polarized distribution pattern of lipids and proteins in the plasma membrane and determines actin cytoskeleton dynamics in the apex, regulating polarized secretion [12,66,67,68]. The use of methyl-β-cyclodextrin, known to extract sterols from membranes and destroy lipid rafts in different cell types [69,70], altered the electrophoretic profiles and ultrastructure of DIMs purified from tobacco pollen tubes [16], suggesting that sterol depletion could also disturb lipid raft nanodomain integrity in pollen tubes.

The goal of this study was to unravel the role of sterols and sphingolipids in tip growth of pollen tubes of *Nicotiana tabacum*. The use of inhibitors of sterol (squalestatin—Sq) and sphingolipid (myriocin—Myr) biosynthesis allowed us to highlight the role of these lipids in linking AF dynamics to the secretion of cell wall/PM components at the tip. The preservation of sterol/sphingolipid ratio proved necessary to maintain a clear zone morphology. Thus, sterol/sphingolipid ratio emerged as a new player linking apical processes involved in pollen tube tip growth.

## 2. Results

### 2.1. Squalestatin and Myriocin Alter Lipid Profile and Pollen Tube Length

To find the lowest concentration of Sq that inhibits sterol biosynthesis without stopping pollen tube growth, pollen was germinated in a control medium and in a medium supplemented with 0.5 μM or 1 μM Sq for 2 h (Figure 1A). Analysis of sterol content in the microsomal fraction (P2) revealed that, while 0.5 μM Sq did not lower the sterol content, a significant decrease in total sterols was detected with 1 μM Sq (Figure 1A).

As total sterol content decreased after 2 h of incubation with 1 μM Sq, these conditions were used in later experiments.

We also investigated the effect of Sq on the synthesis of cycloeucalenol. Microsomes purified from control and treated pollen tubes were analyzed using gas chromatography-mass spectrometry (GC-MS); this analysis showed that the amount of cycloeucalenol also decreased in pollen tubes incubated for 2 h with 1 μM Sq, with respect to controls (Figure 1B).

Myr was used at 5 μM, a concentration which inhibits ceramide biosynthesis without affecting polypeptidic profile (see below). Lipid analysis showed that 5 μM Myr induced a decrease in the content of glycosyl inositol phospho ceramides (GIPC) and glucosyl ceramides (GluCer) compared to control (Figure 1C). Interestingly, a parallel increase in steryl-glucoside (SG) and free sterols was also detected in Myr-treated cells (Figure 1C), suggesting activation of a compensation mechanism.

Sterol levels in the pollen tube membrane were affected by Sq and Myr, as shown by filipin staining (Appendix A). After sterol depletion, the fluorescence in the apex of pollen tubes decreased significantly with respect to control (Appendix A; ANOVA and Tukey’s post hoc test: *p* < 0.001), whereas a significant increase in fluorescence was detected after Myr treatment (Appendix A; ANOVA and Tukey’s post hoc test: *p* < 0.001), in line with lipid content determined in the microsomal fraction.

To investigate whether sterol or sphingolipid depletion affects pollen tube growth, the morphology and length of control and treated pollen tubes were measured after 2 and 2.5 h, without and with Sq or Myr, respectively. Regarding pollen tube morphology, no loss of polarity was observed in drug-treated pollen tubes with respect to control tubes (Figure 1E–G for control, Myr- and Sq-treated pollen tubes, respectively). Nevertheless, pollen tubes treated with 1 μM Sq or 5 μM Myr were significantly shorter than control tubes in three different experiments (Figure 1D; ANOVA and Tukey’s post hoc test: *p* < 0.001), suggesting that lipid perturbation affects pollen tube growth. The same amount of ethanol (Sq solvent) and methanol (Myr solvent) was added to culture media during pollen tube growth in order to analyze if they affected pollen tube growth. No alteration of pollen tube length was observed (data not shown).

### 2.2. Squalestatin and Myriocin Alter the Morphology and Dynamics of the Actin Fringe

Since actin is a major player in pollen tube growth, we decided to explore whether changes in pollen tube elongation involve interaction of actin with cell membranes and the organization and dynamics of the actin fringe.

To investigate whether Sq or Myr modify actin partitioning between membranes (P2) and soluble fraction (S2), P2 and S2 from control and treated pollen tubes were assayed with an anti-actin monoclonal antibody. While the pattern of polypeptides in treated pollen tubes did not seem to differ in P2 and S2 compared to control (Figure 2A,B), western blot analysis showed that both Sq and Myr enhanced the interaction of actin with microsomes, since the amount of actin associated with P2 significantly increased (Figure 2C,D; ANOVA and Tukey’s post hoc test: *p* < 0.001).

In light of these findings, we transformed pollen grains transiently using a plasmid which encodes the fusion protein Lifeact-EGFP (kindly donated by Prof. P.K. Hepler, USA) under the control of the pollen-specific promoter LAT52 (Figure 3A,B).

At the plasmid concentration used, pollen tubes overexpressing Lifeact-EGFP were occasionally observed (about 30% in control and treated samples; Appendix A).

Actin filaments lost their typical distribution and seemed randomly organized in short bundles in the shank of pollen tubes. Since tip growth stopped in these pollen tubes, they were excluded from the analysis.

Distinct populations of AFs were identified along control pollen tubes: axial actin bundles in the shank and distal regions were replaced in the apical flanks (2–5 μm from the apical PM) by short dynamic AFs organized in a fringe (Figure 3A,B; arrow). After Myr and Sq treatment, the long filaments in the shank/distal areas did not show any change in arrangement and only actin fringe organization was modified with respect to control (Figure 3C). Whereas AFs of the fringe appeared more widely distributed in Myr-treated pollen tubes than in control (Figure 3C, compare control and Myr panels), the fringe seemed denser in pollen tubes incubated with Sq, suggesting an increase in the AFs numbers, while the localization of the fringe was similar to that of control cells (Figure 3C, compare control and Sq panels). Similar alterations were never observed in control samples using the same concentration of plasmid.

Quantitative analysis in time-lapse experiments revealed that the mean area occupied by the actin fringe was significantly greater in Myr and Sq than in control pollen tubes (Figure 3D, ANOVA and Tukey’s post hoc test: *p* < 0.001 for Myr and *p* = 0.0055 for Sq). These findings suggest that sphingolipid and sterol depletion modified fringe organization by increasing the number of AFs or by altering their distribution in the apical dome. Moreover, comparison of the mean area occupied by the actin fringe over the course of time showed that differences were significant at each time point (Figure 3E), confirming changes in fringe dynamics in pollen tubes incubated with drugs with respect to control tubes.

### 2.3. Sterols and Sphingolipids Regulate Cell Wall Deposition

Alteration of actin fringe dynamics suggested that Myr and Sq treatment could affect membrane trafficking in the apex, which is involved in cell wall building/modulation. To test the effect of sterol and sphingolipid modifications on cell wall building, the different components of the cell wall were detected with specific probes or immunofluorescence assays on control and Sq- or My-treated pollen tubes.

#### 2.3.1. Pectin Deposition

To reveal pectin secretion, immunofluorescence assays were performed with the monoclonal antibody JIM7 that recognizes highly esterified pectins (HEPs). According to previous observations [71], HEPs were more concentrated in the tip regions than the shank in control cells and cells incubated with the drugs (Figure 4A–F).

Quantification of fluorescent intensity in the tip and shank (Figure 4G,I; white and yellow ROIs, respectively, in Figure 4A) showed that Myr and Sq both induced an increase in esterified pectins in the tip. (tip ANOVA and Tukey’s post hoc test: *p* < 0.0001 for Myr and *p* < 0.01 for Sq) suggesting that modification of the lipid profile could promote SV targeting of/fusion with the tip PM with respect to control tubes. In the shank, the intensification of HEPs fluorescence was significant only for Myr-treated pollen tubes (Shank ANOVA and Tukey’s post hoc test: *p* < 0.005 for Myr and *p* > 0.05 for Sq). Moreover, analysis of the ratio of fluorescent intensity of tip versus shank showed that the difference in esterified pectins between these regions is maintained in pollen tubes incubated with Myr and Sq compared to control (Figure 4H,J; not significant, ANOVA and Tukey’s post hoc test: *p* > 0.05), suggesting that the increase in secretion of HEPs could be accompanied by a concomitant increase in secretion or activity of PME.

Pectin methylesterase activity allows low esterified pectins (LEPs) to accumulate in the cell wall of the shank and distal area. In order to localize LEPs, the monoclonal antibody LM19 was used (Appendix A), showing that LEPs were distributed mainly in the shank and distal area of control pollen tubes as expected, while weak fluorescence was observed in the apex (Appendix A).

The same distribution was maintained after sterol depletion (Appendix A), while in Myr-treated pollen tubes the apex was more fluorescent (Appendix A). Unexpectedly, quantitative analysis of mean fluorescence in the tip and shank revealed a non-significant decrease in LEPs in the shank, and a slight increase in LEPs in the tip after Myr treatment (Appendix A; ANOVA and Tukey’s post hoc test: *p* > 0.01). The decrease in LEPs in the shank and their increase in the apex induced a non-significant increase in the tip/shank ratio of Myr-treated pollen tubes, while Sq-treated pollen tubes were more similar to control (Appendix A). This data suggest that, although PME activity was maintained, as revealed by the tip/shank ratio of HEPs (Figure 4H,J; ANOVA and Tukey’s post hoc test: *p* > 0.01), the increase in HEPs was not accompanied by an increase in LEPs.

Since the increase in pectins could impact cell wall building/properties, the thickness of the cell wall at the apex was measured over time in control and Sq/Myr-treated pollen tubes (Figure 5C). Quantitative analysis showed that the trend of cell wall thickness did not vary significantly between control and treated pollen tubes, suggesting that a portion of the pectins secreted could be internalized (Figure 5C).

As already observed [72,73], tobacco pollen tubes show distinctive pulsed growth with alternating steps of vesicle fusion and tip expansion with a typical periodicity (Figure 5A,B, red and blue line, respectively). Since pollen tube elongation was preceded by an increase in cell wall thickness and the amount of secreted material predicted the growth rate [73], the speed of pollen elongation was measured during pulsed growth (reported as velocity; Figure 5D). The mean velocity did not differ significantly in treated and control pollen tubes, but control pollen tubes achieved higher velocities (up to 0.9 µm/s; Figure 5D) than Myr- and Sq-treated pollen tubes, where lipid perturbation seemed to limit the range of velocities (up to 0.5 µm/s; Figure 5D), suggesting a slowdown of pollen tube growth. In addition, velocity autocorrelation analysis showed up to four peaks in control pollen tubes, while in Myr- and Sq-treated tubes, the curves lost periodicity right after the main peak (Figure 5E), sustaining the idea that oscillations in Sq/Myr-treated pollen tubes are noisier and less periodic than those in controls. Furthermore, cross correlation analysis between thickness and velocity showed that the first peaks occurred earlier in treated tubes than controls (Figure 5F), suggesting that treated tubes grew sooner after wall thickening and that elongation events started earlier with respect to secretion. In conclusion, the Sq/Myr treatments did not have a big impact on overall wall thickness and growth rate, but rather on growth dynamics, showing a quicker response and fewer periodic oscillations.

These differences in pollen tube elongation could be due to differences in cell wall rigidity/extensibility, which depend on PME de-esterification activity. In order to reveal the effect of lipid perturbation on PME secretion, pollen grains were transiently transformed using a plasmid which encodes the fusion protein NtPPME1-EGFP under control of the pollen-specific promoter LAT52 (kindly donated by Prof. M. Bosch, UK) (Figure 6).

To detect fluorescence, we used high plasmid concentrations [71]. NtPPME1 localization in pollen tubes matched that already described in the literature [71] and changes in fluorescence pattern were attributed to the drug treatments. In control pollen tubes, NtPPME1 concentrated in the apical dome of the cell wall and also in unidentified organelles (Figure 6A–C). A similar distribution was observed after Myr treatment (Figure 6D–F), whereas in Sq-treated pollen tubes, a broader distribution of NtPPME1-positive vesicles was observed in the apex (Figure 6G–I). Quantitative analysis performed as a plot profile (Figure 6J) in the apex of pollen tubes (starting 15 µm from the tip) using a freehand line (Figure 6C, blue line) confirmed that NtPPME1 localized in the apical dome and the distribution of protein changed in treated tubes with respect to controls. The parameters of a non-linear mixed model fitting a Gaussian function to the data showed that both the Myr and Sq peaks were significantly narrower than the control one (*p* < 0.05), suggesting that the distribution of NtPPME1was limited to a more restricted area of the apical dome (Figure 6J; compare blue and red lines). In addition, the Myr and Sq treatments both induced a sharper rise of the peak, which was particularly large for the Sq treatment (Figure 6J; compare blue and green lines, *p* < 0.01), confirming accumulation of NtPPME1-positive vesicles in the pollen tube tip. The mean fluorescence of the shanks measured in the plot profile was significantly lower in Myr- and significantly higher in Sq-treated pollen tubes than in controls (Figure 6J; ANOVA and Tukey’s post hoc test: *p* < 0.001). These findings suggest that lipid perturbation affected NtPPME1 secretion or activation.

#### 2.3.2. Cellulose and Callose Deposition

To analyze the effect of lipid modification on cellulose deposition, pollen tubes grown in control medium and in medium containing Myr or Sq were stained using the fluorescent dye GFP-CBM3A, specific for crystalline cellulose (Figure 7).

The distribution of crystalline cellulose appeared similar in control and treated pollen tubes: GFP-CBM3A fluorescence was very low in the apex and crystalline cellulose was detected in the shank/distal area, confirming that cellulose deposition is more active in the latter rather than in the apex (Figure 7A–I). Fluorescent intensity was measured in the cell wall of the tip (Figure 7C, freehand profile up to 5 μm from the tip PM) and shank/distal areas (Figure 7C, freehand profile, from 5 μm to 15 μm from the tip PM). Fluorescent intensity was higher in the shank than in the apex in control, Myr and Sq groups (Figure 7J,L; ANOVA and Tukey’s post hoc test: *p* < 0.01). Comparing control and treated cells, no significant variation in fluorescent intensity with respect to control cells was observed in any region of pollen tubes germinated with Myr or Sq (Figure 7J,L; ANOVA and Tukey’s post hoc test: *p* > 0.05). Consequently, the ratio between tip and shank did not differ between controls and treated pollen tubes (Figure 7K,M), suggesting that lipid perturbation did not affect crystalline cellulose localization.

Callose deposition was detected in control pollen tubes and pollen tubes grown with Myr or Sq using the callose-specific fluorescent dye aniline blue. Just as for cellulose, the deposition of callose did not occur at the very tip but rather in the shank and the distal regions up to about 5 μm behind the tip PM (Figure 8A–I).

Analysis of fluorescent intensity, measured in the apex (Figure 8C; freehand profile up to 5 μm from the tip PM) and in the shank (Figure 8C; freehand profile from 5 μm to 15 μm from the tip PM), showed that callose deposition at the tip of Myr- or Sq-treated pollen tubes did not differ significantly from that of control tubes (Figure 8J,L; ANOVA and Tukey’s post hoc test: *p* > 0.05), suggesting that CalS activity does not start before. Conversely, in the shank of Myr- and Sq-treated tubes, fluorescence intensity was significantly different from that of controls (Figure 8J,L); Myr induced a significant reduction in fluorescence (Figure 8J: ANOVA and Tukey’s post hoc test: *p* < 0.01), which resulted in a significant reduction in the tip/shank ratio (Figure 8K; ANOVA and Tukey’s post hoc test: *p* < 0.001), while in Sq-treated pollen tubes, aniline blue fluorescence increased significantly (Figure 8L: ANOVA and Tukey’s post hoc test: *p* < 0.01). In the latter, the tip/shank ratio did not differ from control, since the increase in fluorescence in the shank was accompanied by a non-significant increase in fluorescence in the tip (Figure 8M). All these findings suggest a modification in callose synthase secretion/activity after lipid perturbation.

Crystalline cellulose and callose deposition from the very tip to 30 μm from the tip PM was studied in more detail using ImageJ-Plot Profile (see plot of mean fluorescent intensity in Figure 9).

A four-parameter logistic function was fitted to the fluorescence data using a non-linear mixed model. Myriocin and Sq mostly seemed to affect cellulose deposition in the distal area of pollen tubes, where GFP-CBM3A fluorescence was significantly lower in treated pollen tubes than controls (Figure 9A; *p* < 0.01). On the contrary, callose deposition appeared more affected in the shank area. In fact, in pollen tubes grown with Myr, the curve was flatter than that of the control, up to about 10 μm from the tip PM, suggesting that callose deposition occurs farther from the tip (Figure 9B). This is also indicated by a significantly lower value of the scale parameter of the logistic curve fitted to the Myr group than to the control group (*p* < 0.01), which indicates an initially flatter curve followed by a steeper increase. In contrast, in Sq-treated pollen tubes the fluorescent intensity grew more quickly toward the distal area, as indicated by the fact that with respect to the control, the inflection point of the fitted logistic curve of the Sq-treated group was shifted towards the tip (Figure 9B; *p* < 0.001). These findings confirm that the secretion/activity of CalS and CSCs were affected by sterol and sphingolipid perturbation.

### 2.4. Investigating the Morphology and Dynamics of the Clear Zone

To determine the role played by sterols and sphingolipids in secretion/sorting, we transiently transformed pollen tubes with LAT52-GFPRabA4d (kindly donated by Prof. E Nielsen, USA), which is specifically expressed in vesicles accumulating in the clear zone, suggesting that it may play a role in vesicle targeting/delivery and fusion at the tip PM and in endocytic vesicle trafficking/recycling in the apical region [74]. In control pollen tubes, the pattern of RabA4d distribution appeared as already reported [75,76]. At the plasmid concentration used, RabA4d did not overexpress and changes in fluorescence pattern were attributed to the drug treatments. As expected, in most transiently transformed LAT52-GFPRabA4d tobacco pollen tubes (about 90%; control), RabA4d localized on vesicles accumulating in the clear zone, periodically with a “tail” extending into the shank, creating the typical V-shape morphology (Figure 10A–C).

Perturbation of lipid rafts by 5 μM Myr or 1 μM Sq altered the distribution of vesicles in the tip (Figure 10D–I, respectively). The sphingolipid depletion/sterol increase triggered by Myr seemed to reduce the area of the clear zone, as the vesicles appeared to concentrate more in the tip dome, and RabA4d-vesicle distribution in the shank appeared less defined than in control pollen tubes (Figure 10 compare A–C and D–F; about 85% of pollen tubes). This morphological alteration was accompanied by a significant reduction in mean fluorescent intensity (Figure 10M; ANOVA and Tukey’s post hoc test: *p* < 0.001) and a significant increase in the mean area (%) occupied by RabA4d compartments with respect to control tubes (Figure 10J,K,N; ANOVA and Tukey’s post hoc test: *p* < 0.01). Treatment with Sq induced a more severe modification of the distribution of RabA4d-vesicles. Pollen tubes with a reduced quantity of sterols showed expansion of the clear zone with the V region extending more deeply from the tip into shank/distal regions (Figure 10 compare A–C and G–I; about 65% of pollen tubes). Similar alterations were never observed in control samples at the amount of plasmid used. Analysis of fluorescent intensity in the apex did not reveal differences between control and Sq-treated pollen tubes. On the contrary, a significant increase in mean fluorescent intensity and an increase in the area occupied by RabA4d compartments with respect to control were detected in the shank of sterol-depleted pollen tubes (Figure 10J,L–N; ANOVA and Tukey’s post hoc test: *p* < 0.01).

As a whole, these findings suggest that lipid raft perturbation severely alters the morphology of RabA4d compartments and affects membrane trafficking in the clear zone.

## 3. Discussion

Cell polarity and tip growth are involved in a wide range of biological processes in different organisms. The growing pollen tube is a fascinating model for studying the mechanisms underlying polarized growth. Besides the asymmetric localization of proteins along the PM and the unequal distribution of organelles in the cytoplasm, there has been recent evidence of a polarized distribution of sterols and liquid-ordered domains in tobacco pollen tubes [16].

In the present study, which used an experimental approach involving the perturbation of membrane lipids through inhibition of sterol and sphingolipid biosynthesis, we gained more insights into the role of these lipids during pollen tube growth in *Nicotiana tabacum*. The use of inhibitors is a useful and accepted method of addressing the question of the role of sterols and sphingolipids in membrane formation and function [30]; moreover, by adjusting drug concentration, we were able to modulate the effect of lipid depletion on membrane function.

The impact of sterol perturbation on lipid microdomains was demonstrated by treating the PM of BY2 cells with β-cyclodextrin, which disrupted phytosterol-rich lipid-ordered domains [70]. In particular, β-cyclodextrin also affected the ultrastructure and composition of DIMs isolated from pollen tubes [16]. This work shows that sterol depletion influences processes occurring mostly in the apex of pollen tubes, where lipid-ordered domains and sterols concentrate. Considering all the results, we concluded that variations in the sphingolipid/sterol ratio mainly impacted microdomains, revealing their role in pollen tube growth. However, we cannot exclude the possibility that variations in these lipids influence membrane properties, independently of lipid microdomain formation/function.

To inhibit sterol and sphingolipid biosynthesis, we used Sq and Myr, respectively, at the lowest concentrations that inhibited lipid biosynthesis without affecting polypeptide profile: 5 μM Myr and 1 μM Sq, which allowed pollen tubes to grow and to maintain a polar distribution of organelles. Consistently, 5 μM Myr has been shown to deplete ceramides in *Chlamydomonas reinhardtii* [77], while a recent study by Villette et al. [78] showed that incubation with 1 μM Sq induced sterol depletion in tobacco pollen tubes.

Alteration of the sterol/sphingolipid content of cell membranes had pleiotropic effects; it affected actin fringe morphology/dynamics and changed how much actin interacted with cell membranes, having a synergic effect on secretory/recycling activity in the apex and suggesting that sterols and sphingolipids have a role in regulating/modulating the deposition of cell wall components.

### 3.1. Sterols and Sphingolipids Play a Role in Modulating Actin Filament Organization and Dynamics in the Apical Region and in Defining Pollen Tube Pulsed Growth

Analysis of lipid biosynthesis during microsporogenesis has shown that pollen grains contain a stock of sterols at anthesis [78,79]. However, de novo synthesis of cycloeucalenol has been described during pollen germination and pollen tube growth in tobacco [78,79]. In our study, 1 μM Sq induced sterol depletion in tobacco pollen tubes and the decrease in sterols was mostly due to inhibition of cycloeucalenol biosynthesis. On the contrary, the depletion of GIPC and GluCer induced by 5 μM Myr occurred in parallel with the increase in total steryl-glucosides (SG) and free sterols, probably to compensate for the decrease in membrane sphingolipids. In addition, the inhibition of sphingolipid synthesis induces a shift in the utilization of glucose (less GluCer and more SG). Since sterols are necessary to promote liquid-ordered domains [30,80], we may hypothesize that sterol deprivation affects association of lipids in microdomains in pollen tubes as well. On the contrary, sphingolipids alone are unable to promote formation of the liquid-ordered phase but can increase the number of microdomains in the presence of sterols [81]. This suggests that after Myr treatment, different microdomains enriched in free sterols and SGs, with different properties and functions, may be formed. Although both drugs were used at the lowest concentration that alters sterols and sphingolipids without affecting pollen tube morphology, retarded pollen tube growth was observed, suggesting that sterols and sphingolipids play a role in apical elongation processes.

One of the major players in pollen tube growth is the cytoskeleton. Both tubulin and actin have been demonstrated in DIMs, suggesting that they may participate in regulating the function of AFs and MTs in pollen tubes [16]. This is consistent with the increase in the amount of actin associated with membranes and with changes in the dynamics and organization of the actin fringe observed after perturbation of membrane lipids with Myr and Sq. It is not known whether actin is associated with membranes in its monomeric form or as filaments interacting with proteins involved in its nucleation/polymerization. Live-cell imaging experiments have revealed that actin nucleation occurs continuously at the PM of the pollen tube tip and that this actin polymerization is required for pollen tube growth [82]. It is therefore possible that actin associates with membranes as filaments which are continuously nucleated from the PM in growing tips [82,83,84]. In fact, actin monomers distribute uniformly in the cytoplasm of Arabidopsis pollen tubes and are rapidly redistributed via cytoplasmic streaming to polymerization sites [85]. Alteration of actin partitioning between P2 and S2 after perturbation of lipids with Myr and Sq in tobacco pollen tubes suggests that sterols and sphingolipids may be involved in fine-tuning actin nucleation/remodeling during pollen tube growth, possibly by modulating the activity of tip-localized proteins.

It was recently reported that two distinct populations of AFs are polymerized from the PM in Arabidopsis, Lily and tobacco pollen tube tips. In the apical flanks, cortical AFs are organized in a fringe that regulates targeting of SVs for exocytosis, while apical AFs extend into the inner region of the cytoplasm, acting as a physical barrier to backward movement of vesicles [82,83,84,86]. The idea that sterols and sphingolipids in liquid-ordered apical domains are involved in cytoskeletal organization of the pollen tube apex is confirmed by the alteration of actin fringe organization/dynamics in tobacco pollen tubes after lipid perturbation. In Myr- and Sq-treated pollen tubes, the actin fringe was more persistent and did not show the usual pulsed dynamics. A more stable actin fringe induced by Sq/Myr could be responsible for maintaining more post-Golgi SVs in the apical region, thus increasing their chance of interacting with the tip PM [84].

### 3.2. Pollen Tube Cell Wall Building Requires Integrity of Sterol/Sphingolipid or Ganization in Membranes

Data derived from inhibition of sterol and sphingolipid biosynthesis may imply changes in the dynamics of the secretory pathway, leading to alterations in cell wall composition and stiffness and/or changes in turgor pressure, due to Ca^2+^ and proton fluxes [50]. A relation between the equilibrium of ceramides in membranes and cell wall stiffness/turgor pressure has been shown during Arabidopsis pollen tube growth [87]. We observed a similar effect in the tobacco pollen tube cell wall after Myr and Sq treatments. Sterol and sphingolipid perturbation induced an increase in pectin secretion, which was not, however, accompanied by an increase in cell wall thickness. It is possible to suppose that lipid perturbation altered the balance between cell wall stiffness and turgor pressure, affecting pollen tube growth. In fact, while a non-significant slowdown in growth rate was detected in control and Sq/Myr-treated pollen tubes, autocorrelation and cross-correlation analysis between cell wall thickness and velocity suggested that lipid perturbation induced an alteration in secretion/elongation phase oscillations. In tobacco pollen tubes, the growth process is the result of the time pollen tubes spend in elongation phase added to the time spent in cell wall building [50]. Lipid perturbation disturbed the success of oscillatory growth since the time between cell wall thickening and distension was shorter than in controls. Interestingly, this change in growth mechanisms did not influence the pollen tube growth rate as a whole. Consequently, lipid perturbation could modify cell wall deposition, upsetting the interval between cell wall stiffening and the increase in turgor pressure. On the other hand, the shorter pollen tubes observed after Myr and Sq treatment could be mainly due to alteration of processes occurring in the early phases of tube emission (manuscript in preparation).

Pollen tube oscillatory growth in tobacco depends on the pulsed exocytosis preceding elongation, while growth rate depends on the amount of cell wall material secreted [72,73]. Furthermore, LEPs in the shank/distal region contribute to cell wall stiffness while HEPs in the apex allow apical distension. Different mechanisms have been proposed to explain how cell wall and turgor pressure work together for proper apical growth [73,88,89,90,91], although it is known that HEPs in the tip region provide sufficient elasticity to promote pollen tube growth [92,93]. In our study, changes in the correlation between velocity of elongation and cell wall thickening in tobacco pollen tubes could also be due to alteration of pectin secretion and pectin de-esterification. The increase in HEPs detected in the apex and shank, suggested a rise in pectin secretion in the tip region.

A balance between methylated and demethylated pectin levels is critical for proper pollen tube growth. Maintenance of the tip/shank fluorescence ratio in treated and control tobacco pollen tubes suggested that, although abundant, HEPs were normally de-esterified by NtPPME1. Nevertheless, analysis of LEPs showed that the increase in HEPs was not accompanied by an increase in LEPs in the tip and shank. On the contrary, although the fluorescence of LM19 antibody was lower in Sq- and Myr-treated cells than in control pollen tubes, this difference was not significant. As clathrin-mediated endocytosis (CME) is selectively involved in the internalization of demethylated pectin [94,95,96,97], the decrease in LEPs in the shank of tobacco pollen tubes after lipid perturbation could be due to enhanced internalization of these pectins, allowing cell wall thickness to remain as in control pollen tubes. In previous studies, charged nanogold was used to show that CME occurs at the tip and in subapical regions of tobacco and Arabidopsis pollen tubes [64,98]. In tobacco pollen tubes, LM19 staining suggested that lipid perturbation could affect the endocytic processes in the shank. These findings support the idea that the increase in HEPs was mostly due to an increase in pectin secretion at the pollen tube apex, coupled with an increase in internalization of LEPs in the shank. The puzzle is complicated because phosphoinositides, such as PtdIns(4,5)P_2_, play a role in the control of CME in pollen tubes [99]. A pool of PtdIns(4,5)P_2_ was detected in PM domains enriched in sterols and sphingolipids [17]. Perturbation of the sterol/sphingolipid ratio could impact on PtdIns(4,5)P_2_, affecting CME of LEPs in the shank.

Early studies suggested that esterified pectins and PME co-localized in Golgi-derived SVs, supporting the hypothesis that PME could be transferred to the cell wall in a precursor form and be activated at the tip [100,101]. A recent study on *Nicotiana tabacum* pollen-specific pectin methylesterase 1 (NtPPME1) revealed that the polarized exocytosis and apical targeting of NtPPME1 are directly mediated by Golgi-derived vesicles which by-pass the Trans Golgi Network (TGN), thus taking a different pathway with respect to pectins [51,52,102]. Pectin methylesterase 1, used in this study in transiently transformed pLAT52-NtPPME1-GFP tobacco pollen tubes, has a pro-region which is presumed to be an endogenous inhibitor (pro-PME1) [71,103]. The pro-region together with PMEI [103] regulated the activity of PME. This pro-region is cleaved after pro-PME1 secretion, allowing full activity of the enzyme in the shank where the cell wall needs to be stiffened [71,103]. The localization of NtPPME1 in control tobacco pollen tubes was consistent with localization of the enzyme carrying the pro-region [71,73]. In fact, we observed NtPPME1 in the central dome of the tip with a highly focused tip gradient distribution and it did not accumulate in the vesicles of the clear zone. Pectin de-esterification occurred on the shoulders of the pollen tube apex, where NtPPME1 fluorescence was not detected. In tobacco pollen tubes, NtPPME1 distribution appeared to depend on sterol/sphingolipid ratio. In fact, Myr and Sq both induced a reduction in NtPPME1 secretion area at the very tip, while only sterol depletion allowed this enzyme to accumulate in the clear zone vesicles. Sterol/sphingolipid perturbation may induce a double effect. Reduction in the arc of fluorescence decorating the tip in Myr/Sq-treated pollen tubes suggests faster cleavage of pro-region and earlier activation of PME1 than in control cells, supporting the idea that these lipids are involved in fine regulation of NtPPME1 at the pollen tube tip. Thus, PME1 could act earlier in de-esterifying pectins, allowing the tip/shank HEP ratio to be maintained in treated and control pollen tubes. Furthermore, since the pectin and PME secretion paths are different [51,52], sterol- and sphingolipid-enriched domains seem to be involved in modulating TGN-dependent and -independent secretion events. The appearance of NtPPME1-positive vesicles in the clear zone after sterol depletion suggests that secretion or vesicle dynamics in this area may be affected by lipid perturbation (see below).

Since cell wall stiffness is also determined by cellulose and callose in the shank/distal areas of the pollen tube [92,104], the nature of cell wall modification induced by lipid perturbation was investigated by analyzing cell wall components using specific fluorescent probes for crystalline cellulose and callose. In the shank and distal regions, these wall components are highly resistant to tensile stress [54], while the non-crystalline state of cellulose microfibrils at the apex ensures that they do not impair pollen tube distension [54,92]. Nevertheless, plot profile analysis suggested that in tobacco pollen tubes, cellulose and callose deposition were, in any case, affected by lipid perturbation. Specifically, crystalline cellulose deposition decreased in the shank, suggesting that CSC was ineffective. This modified the nature of cellulose which in turn was unable to assemble into crystalline structure. Different studies on pollen tubes have revealed a special mechanism for cellulose deposition in these cells. In fact, cellulose appeared as a heteropolymer containing β-1,4 links and a few β-1,3 links, rather than a homogeneous β-1,4-linked glucan [2,55,105], making it more similar to lower eukaryote cellulose. Furthermore, unlike in somatic cells [106], the distribution of cellulose synthase in pollen tubes seemed to depend on actin filaments and endomembrane dynamics. Alteration of the actin fringe and the concomitant modification of clear zone trafficking and dynamics due to perturbation of the sterol/sphingolipid ratio caused differences in cellulose synthase activity in the shank. Various studies suggest that the lipid environment of CSCs regulates the organization and function of CESA in the PM lipid bilayer [107,108,109,110]. In Arabidopsis mutants, it has also been reported that defects in sterol biosynthesis affect cellulose deposition in somatic cells, as sterols are the first acceptor of glucose (to produce SG) in the nucleation phase during the formation of cellulose polymer by CSCs [56,110]. Changes in sterol content induced by Sq and Myr (decrease and increase in sterols, particularly SG, respectively) could affect nucleation of cellulose microfibrils and therefore wall crystalline cellulose content.

The altered sphingolipid/sterol ratio may also have altered CalS activity along the pollen tube. CalS is secreted in an inactive form and activation occurs on the PM with a partially characterized mechanism [111]. In tobacco pollen tubes, callose deposition was affected by lipid perturbation in the shank, suggesting that sterol/sphingolipid could participate in the activation of this enzyme after its secretion. Interestingly, detergents that alter membrane fluidity and lipid-lipid interaction in membranes modified CalS activity [47], confirming the importance of the lipid environment for enzyme functions. Alternatively, since lipid rafts are responsible for modulating actin fringe dynamics and membrane sorting in post-Golgi secretion, modifications in the lipid profile may alter the dynamics of secretory activity in the tip, leading to alteration of CalS secretion. Notably, the CalS complex is activated by ROP1 which seems to regulate the process of callose synthesis [112,113]. Since ROPs are associated with sterol-enriched microdomains in Arabidopsis somatic cells [114,115,116], sterols and sphingolipids could play a role in callose deposition, supporting CalS and ROPs interplay.

### 3.3. The Morphology and Dynamics of the Clear Zone Depend on Stability of the Sterol/Sphingolipid Ratio

Modification of the actin fringe, the distribution of cell wall components and NtPPME1 trafficking after Sq and Myr treatment suggest that exocytosis events could be affected by sterol/sphingolipid perturbation in the pollen tube apex. Modification of sterol/sphingolipid ratio may influence SV fusion events at the PM and/or the function of membrane trafficking involving organelles such as Golgi and the TGN.

Previous studies with filipin showed that sterols accumulate in the clear zone of tobacco pollen tubes [16]. The origin and maintenance of the clear zone during pollen tube growth, as well as its function, have long been studied [117]. The clear zone was considered to contain SVs, conveyed to the tip region by cytoplasmic streaming along actin bundles which do not invade the tip, while short fine AFs in the apex were involved in trapping SVs, thus favoring their interaction with the apical PM [82,86]. Several studies have suggested a more complex picture, showing that this region also comprises tip-internalized endocytic vesicles and recycling vesicles [64,65,118]. In particular, microtubule (MT) depolymerization experiments with nocodazole allowed us to hypothesize that, in addition to TGN, the clear zone might be a further sorting compartment involved in control of PM composition in the tip region [75]; thus, vesicles in this region could be defined as transport vesicles, rather than simply SVs [74].

To unravel the role played by sterols and sphingolipids in relation to clear zone function, we transiently transformed pollen tubes with LAT52-GFPRabA4d. This protein belongs to the RabA4 subfamily of Rab GTPases, which are key regulators of membrane trafficking: RabA4d is the pollen-specific homologue of RabA4b, which localizes in the TGN and in endosomal vesicles recycling in root hair tips [75,119]. RabA4d is specifically localized in vesicles accumulating in the clear zone of growing pollen tubes, suggesting its involvement in regulating vesicle targeting/delivery and vesicle fusion at the pollen tube tip [74]. It has also been shown that RabA4d co-localizes with FM4-64 in tobacco pollen tubes, suggesting that it could also be involved in endocytic vesicle trafficking/recycling in the apical region [74]. As a whole, the data on tobacco pollen tubes suggests that sterol/sphingolipid perturbation severely alters the morphology of RabA4d compartments, affecting membrane trafficking in the clear zone. The different clear zone organization/dynamic responses depend on sterol or sphingolipid perturbation. Myriocin induced an accumulation of Raba4d-positive vesicles at the very tip and partial disorganization of the V-shaped region, suggesting that changes in the sterol/sphingolipid ratio altered membrane trafficking in the clear zone. In Sq-treated pollen tubes, the clear zone was entirely disorganized, with Raba4d-positive vesicles distributed all along the shank and distal area of pollen tubes, suggesting that the fusion/recycling of transport vesicles was delayed/obstructed at the tip in treated cells. Consequently, more RabA4d vesicles were captured by reverse fountain streaming, becoming widespread in the shank and distal regions. The RabA4d distribution could be due to alteration of AF organization in the pollen tube apex, disrupting the clear zone and altering exocytosis. As an alternative or in parallel, since the effect on clear zone dynamics seemed more severe after sterol depletion, Sq treatment could affect organelles involved in sorting and recycling endocytic vesicles. Trans-Golgi networks may no longer be able to sort and address vesicles to the appropriate targets, leaving them to accumulate in the cell. On the other hand, the lower impact on TGNs and post-Golgi trafficking of Myr could be due to the compensatory effect of sterols. Furthermore, it has been demonstrated that GlcCer biosynthesis is required for the maintenance of Golgi morphology and membrane trafficking/protein secretion [120,121,122,123], explaining the effect of changes to the clear zone in Myr-treated pollen tubes. More experiments analyzing the behavior of TGNs after Sq and Myr treatment could clarify the role played by sterols and sphingolipids in post-Golgi trafficking.

The clear zone could therefore be a hub where different secretory pathways converge, and simultaneously a sorting compartment to direct endocytic and SVs to various destinations. The changes in clear zone morphology and dynamics after lipid perturbation support the idea that sphingolipids and sterols could modulate the sorting function of this compartment.

These alterations affect cell wall building in tobacco pollen tubes, as previously described. Interestingly, Raba4d-null mutants showed that this protein is important for proper trafficking/deposition of pectins to/in the cell wall [74]. In fact, in these mutants, HEPs were observed in the tip and distal regions, indicating that RabA4d plays a role in pectin delivery to the PM. The pattern of pectin distribution observed after lipid perturbation (see above) mimicked what was observed in the Raba4d-null mutant, suggesting that sterols and sphingolipids are involved in RabA4d vesicle trafficking and fusion to the PM. It was also observed that callose distribution in the cell wall was not affected in the Raba4d-null mutant, suggesting that pectin and callose synthase (CalS) secretion have different pathways [74]. Actually, alteration of callose and cellulose deposition in the tube cell wall, induced by Myr and Sq, could be not ascribed to altered exocytosis of these enzymes. However, since both CalS and CESAs have been reported in isolated DIMs [56,60,124], a possibility is that differentially modulated lipid profiles of microdomains enriched in sterols and sphingolipids may regulate the activation of enzymes on the PM.

## 4. Conclusions

The present study shows that sterols and sphingolipids affect AF polymerization/depolymerization in the tip of pollen tubes. Alteration of the actin fringe as a result of lipid perturbation induces changes in the sorting function of the clear zone, which affects the cell wall’s structure and modulates the activity of enzymes involved in cell wall deposition. In turn, cell wall modifications alter the pattern of pollen tube growth, modifying the pauses between the elongation and secretion phases, without affecting the pollen tube growth process as a whole. These findings suggest that sterols and sphingolipids may have an important role not only as fundamental structural components of membranes, but also as an additional player in the coordination and/or interconnection of intracellular membrane processes. In pollen tubes, sterols and sphingolipids play a role in apical growth, finely coupling actin dynamics and vesicle trafficking processes.

How sterol and sphingolipid ratio regulate or promote crosstalk between different cell processes governing pollen tube growth is still unclear. The present findings offer many suggestions, opening intriguing new horizons for future research.

## 5. Materials and Methods

### 5.1. Probes and Drugs

Zaragozic acid/squalestatin (Sigma Aldrich, St. Louis, Missouri, USA) was dissolved in Ethanol to a concentration of 6.6 mM and then diluted to concentrations of 0.5 μM and 1 μM in the culture medium. A stock solution of myriocin (Sigma Aldrich, St. Louis, Missouri, USA) in methanol was prepared to a concentration of 4.98 mM and diluted to 5 μM concentration in the culture medium. GFP-CBM3A (NZYTech, Lisboa, Portugal) and aniline blue (Sigma Aldrich, St. Louis, Missouri, USA) were used to a final concentration of 0.02 mg/mL and 0.1%, respectively. Fresh solution of Filipin (Sigma Aldrich, St. Louis, Missouri, USA) was used at final concentration of 500 μM in DMSO.

### 5.2. Germination Assay and Pollen Tube Measurement

Pollen of *Nicotiana tabacum* (L.) was collected from plants grown in the Botanical Garden (Città Studi) of Milan University during summer and stored at –20 °C. Pollen grains (3 mg/mL) were cultured in flasks, in BK liquid medium [125] supplemented with 12% (*w*/*v*) sucrose at 23 ± 2 °C with or without squalestatin 0.5 μM/1 μM or myriocin 5 μM for 2 h and for 2.5 h, respectively. Control and treated pollen tubes were fixed (fix solution: 12% sucrose, 100 mM PIPES pH 6.9, 5 mM MgSO_4_, 0.5 mM CaCl_2_, 3.7% formaldehyde; squalestatin 0.5 μM/1 μM or myriocin 5 μM were added in fix solution) and observed with the Leica optical microscope DM RB, using a Leica N PLAN 10X objective. Images were collected with the Leica video camera MC 170 HD. The length of control and treated pollen tubes was calculated using ImageJ 1.53t software (National Institutes of Health) and analyzed by ANOVA test.

### 5.3. Pollen Tube Microsomes

*Nicotiana tabacum* (L.) pollen collected in the Botanical Garden Città Studi, as described above, was hydrated in a humid chamber overnight. Pollen (3 mg/mL) was germinated in BK medium as reported above, with or without squalestatin 1 μM or myriocin 5 μM. Pollen tubes were rinsed with 10 mL of incomplete TNE buffer (mM Tris, 150 mM NaCl, mM EGTA, 1 mM PMSF, 10 μg/mL TAME) containing 12% sucrose, with or without squalestatin 1 μM or myriocin 5 μM and centrifuged at 2000 r.p.m. for 10 min at 10 °C in a Beckmann JS13.1 rotor. Pollen tubes were homogenized on ice in two volumes of complete TNE (mM Tris, 150 mM NaCl, mM EGTA, 1 mM PMSF, 10 μg/mL TAME, 10 μg/mL leupeptin, 10 μg/mL pepstatin A, 4 μM aprotinin, 8 μM antipain) using a 2 mL Potter (teflon/glass) homogenizer. The homogenate was centrifuged at 572 g for 4 min at 4 °C and the post nuclear supernatant loaded onto a 20% sucrose cushion (3 mL) in incomplete TNE buffer and centrifuged at 64,200 g (Beckman SW-60 rotor) for 30 min at 4 °C. The P2 pellet was resuspended in cold, complete PEM buffer. Aliquots of P2 and supernatant (S2) were protein-assayed (Bradford) using BSA as standard protein.

### 5.4. Lipid Analysis

For the analysis of sphingolipids and glucosylceramides, 1 mL of hot isopropanol/hexane/water (55/20/25, *v*/*v*) was added in each P2 pellet fraction. The solution was incubated at 60 °C for 20 min. After centrifugation at 3000 g for 10 min, the pellet was extracted twice more with the hot solvent. The supernatants were evaporated until dry at 40 °C and resuspended in tetrahydrofuran/methanol/H_2_O (40/20/40 *v*/*v*). Their lipid content was analysed with High Performance Thin Layer chromatography (HPTLC). Before the analysis, HPTLC plates (Silicagel 60 F254 Merck) were impregnated for 3 min with freshly prepared 0.2 M ammonium acetate in methanol, and further dried at 110 °C for 15 min. The lipid extracts were chromatographed in chloroform/methanol/4 N NH_4_OH (9/7/2, *v*/*v*). For the analysis of sterols, each P2 pellet fraction was resuspended in 0.5 mL H_2_O, then 2 mL chloroform/methanol (2/1, *v*/*v*) was added and the sample was vortexed. After 1 h, the tubes were centrifuged at 3000 g for 10 min and the supernatant was evaporated to dryness. The lipid extracts were analyzed by HPTLC or further processed for GC-MS. For HPTLC, lipids were resuspended in chloroform/methanol (2/1, *v*/*v*) and chromatography was performed using hexane/ethyl ether/acetic acid (90/15/2, *v*/*v*). For GC-MS, saponification was first carried out through addition of 1 mL of ethanol and 0.1 mL of KOH 11 N to each lipid extract as well as 5 μg cholesterol as an internal standard; tubes were incubated at 80 °C for 1 h; sterols were extracted with 1 mL of hexane and 2 mL H_2_O were added; tubes were vortexed and centrifuged at 3000 g for 10 min. The supernatant was transferred to another tube and was evaporated to dryness under nitrogen gas stream before silylation. Sterols were derivatized with 200 μL of N,O-Bis(trimethylsilyl)trifluoroacetamide with 0.1% triméthylchlorosilane (BSTFA/TMCS) and incubation at 110 °C for 15 min. BSTFA was evaporated under nitrogen gas stream, and the samples were dissolved in hexane. GC analysis was carried out with an Agilent 7890A Plus GC unit with flame ionization detector (Agilent, Santa Clara, CA, USA). Silylated sterols were separated on a 30 m HP-5MS column (Agilent) using a temperature gradient of 150 °C increased to 280 °C at 10 °C min^−1^, held for 10 min, and decreased to 150 °C at 20 °C min^−1^. Sterols were identified by GC-MS (Agilent, Santa Clara, CA, USA) using the same column and temperature gradient.

### 5.5. SDS-PAGE, Western Blot

SDS-PAGE analysis was performed using 10% linear acrylamide concentration according to the method of Laemmli [126]. Gels were stained with Coomassie Brilliant Blue R250. Western blot was performed according to Towbin et al. [127]. The anti-actin monoclonal antibody (Sigma, USA) was used at final dilution of 1:25.000 and detected as outlined in the Amersham ECL kit booklet. The signal was acquired using Chemidoc system (BioRad, Hercules, CA, USA). For quantification of protein level, the area of bands was considered. Quantitative analyses were performed with the ImageJ Software and statistically analyzed with ANOVA test.

### 5.6. Immunofluorescence Analysis

Pollen of *Nicotiana tabacum* was germinated with or without 1 μM squalestatin or 5 μM myriocin and fixed as reported above. Cells were rinsed 3 times for 5 min using TBS and then incubated with JIM 7 rat monoclonal antibody or LM19 1: 15 overnight at 4 °C. Cells were rinsed in TBS and later incubated with anti-rat FITC antibody 1:200 for 2 h. After three rinse in TBS, samples were observed using a Leica TCS SP8 microscope with a 63X oil immersion (NA 1.4) objective (Leica Microsystems, GmbH, Wetzlar, Germany). The 488 nm laser lines were used and fluorescence was collected in the 480–520 nm emission window. The acquisition parameters were maintained for each experiment so that the images were comparable. Mean fluorescence intensity in tip and shank was measured using ImageJ software and statistically analyzed with ANOVA test.

### 5.7. GFP-CBM3A and Aniline Blue Staining

Pollen tubes grown in BK medium, 12% sucrose with and without 1 μM squalestatin or 5 μM myriocin, were fixed as reported above before staining. For GFP-CBM3A, staining cells were rinsed 3 time for 5 min in PBS solution and then incubated with GFP-CBM3A solution 20 µg/mL final concentration for 45 min. After a rinse in PBS, samples were observed with Leica TCS SP5 confocal microscope with a 63x oil immersion (NA 1.4) objective (Leica Microsystems, GmbH, Wetzlar, Germany). The 488 nm laser lines were used to excite GFP and fluorescence was collected in the 500–550 nm emission windows. For Aniline, blue staining cells were rinsed 3 time for 5 min in TBS solution and callose was visualized using 0.1% aniline blue in K_3_PO_4_. The 405 laser line were used to excite Aniline blue and the fluorescence was collected in the 460–550 nm emission windows. Mean fluorescence intensity in tip and shank was performed with a 20 points freehand line and analyzed as reported below.

### 5.8. Filipin Staining

Pollen tubes grown as previously reported were fixed in formaldehyde 3.7% in PM buffer (Pipes 0.1 M, MgSO_4_ 5 mM, Ca Cl_2_ 0.5 mM, sucrose 10%) for 40 min. After three rinses in PBS, pollen tubes were incubated with filipin 500 µM in DMSO, for 2 h at room temperature and then overnight at 4 °C. Pollen tubes were rinsed in PBS and images were obtained with a Leica MC170HD camera mounted on a Leica fluorescence microscope DM RB (Leica Germany), equipped with an A combination filter. Mean fluorescence intensity was measured using ImageJ software and statistically analyzed with ANOVA test.

### 5.9. Transient Gene Expression

For transient gene expression, pollen grains were collected from fresh flowers of *Nicotiana tabacum* (L.) and allowed to germinate at 23 °C on solid medium as reported by Kost et al. [128]. The expression vectors pLIFEACT-EGFP, pLAT52- NtPPME1-GFP and pRABA4D-GFP were transferred to mature pollen grains on solid culture medium using a helium-driven particle accelerator (PDS-1000/He; Bio-Rad, Hercules, CA, USA). Pollen grains were placed under the stopping screen at a distance of 8 cm and bombarded in a vacuum of 28 inches of mercury using a helium pressure of 1100 psi, according to the manufacturer’s recommendation (Bio-Rad, Hercules, CA, USA). Gold particles (1 μm) were coated with plasmid DNA. We tested increasing plasmid concentrations and the lowest concentration at which staining could be observed was chosen. pLIFEACT-EGFP (4 μg), pRABA4D-GFP (4 μg), or pLAT52-NtPPME1-GFP (15 μg) were used to coat 1.5 mg of gold particles. In order to preserve the expression level, the same amount of cDNA-encoding protein was loaded on the gold particles in control and treated pollen tubes. There were no differences in control and treated pollen tubes due to different protein expression, since pollen tubes with similar fluorescent intensity were considered during acquisition. Bombarded cells were kept at 23 °C in the dark for 5 h before observation and samples were incubated with PTNT medium with or without 1 μM squalestatin or 5 μM myriocin for 2 h and 2.5 h before observation, respectively. Observations were performed using a Leica TCS SP5 microscope with a 63x oil immersion (NA 1.4) objective (Leica Microsystems, GmbH, Wetzlar, Germany). The 458 nm laser lines were used to excite YFP and fluorescence was collected in the 520–550 nm emission windows. Control and squalestatin- or myriocin-treated pollen tubes transformed with pLIFEACT-EGFP or pRABA4D-GFP were observed using a SP2 CLSM (Leica Microsystems, GmbH, Wetzlar, Germany) equipped with an argon ion laser. For time-lapse experiments, EGFP and GFP were excited using the 488 nm laser line and fluorescence was imaged between 480–520 nm. A 63X Leica oil immersion plan apo (NA 1.25) objective and a 2.0 zoom lens were used for all the experiments. A minimum of three experiments was performed for each treatment. To compare different experimental conditions, live data mode acquisitions were always performed with the same laser intensity and PMT settings. Time course analysis was carried out with the Leica TCS SP2 software time course option (15 frames—minimize) for 250 s. Images acquired by confocal microscopy were deconvolved with Huygens Professional version 19.04 (Scientific Volume Imaging, The Netherlands, http://svi.nl; accessed on 13 April 2021) and loaded into the open-source software program CellProfiler (Broad Institute, Cambridge, MA, USA). Images were then segmented to identify objects. The process had four main steps: identification of the tube and rotation in order to align all of them vertically, creation of a ROIs of 5 μm from the tip (pLIFEACT-EGFP), a ROI of 5 μm from the tip and a ROI of 60 μm for shank and distal region (pRABA4D-GFP), identification of staining (green channel) and tabulation of measurements. Upon completion of the pipeline, a spreadsheet containing the percentage of area covered by MFs or RabA4d over total area of the tube in the ROI for each tube was generated and exported for further image analysis. Values were processed for statistical analysis using ANOVA test.

### 5.10. Tube Growth Data Analysis

Time lapses of pollen tube growth were analyzed with a custom-built Matlab code (See Appendix A for a detailed description). Briefly, the growth trajectory is identified first as the central line of the time projection of the tube segmentation at all the times. Then, the tube is more finely segmented for all the time frames, and, at the intersection of the segmentation mask with the growth trajectory, is used as reference for the identification of the tube tip. The exact width and middle position of the tube tip were measured, fitting the intensity profile with a Gaussian. The measures are taken as the max position of the Gaussian and its standard deviation. The intensity profile is mediated over a width of three pixels. To calculate the velocity and thickness correlations, the data are first cleaned of some noise by applying a running average over 11 time-points and then an overall baseline is subtracted. For the accurate description of image analyses see Appendix A.

### 5.11. Statistical Analysis of Fluorescence and Profile Variation of NtPPME1, GFP-CBM3A and Aniline Blue

Statistical analyses were run using mixed models and non-linear mixed models with R program. The first ones were used to compare mean fluorescence values between treatments, between tip and shank of the tube, and according to their interaction (if significant). In these analyses, the tube ID was entered as a random factor to account for the repeated measures of the tip and shank of the same tube. Post hoc comparisons between treatments and control groups were performed with the Dunnett’s method. Non-linear variation in the fluorescence along the profile of the pollen tubes were fitted with non-linear mixed models where we entered the tube ID as a random factor and allowed the parameters of the fitted curve to vary between treatments. In the analyses of NtPPME1, GFP-CBM3A and aniline blue we fitted, respectively, a Gaussian, a four-parameter logistic and a three-parameter logistic function. Full details of the analyses are reported in the Appendix A. Models were fitted with the lme and nlme procedures in the nlme package of R 4.0.5 (R Core Team 2021). Post hoc comparisons were run with the emmean procedure of R 4.0.5. For the accurate description of statistical analyses see Appendix A.

## Figures and Tables

**Figure 1 plants-12-00008-f001:**
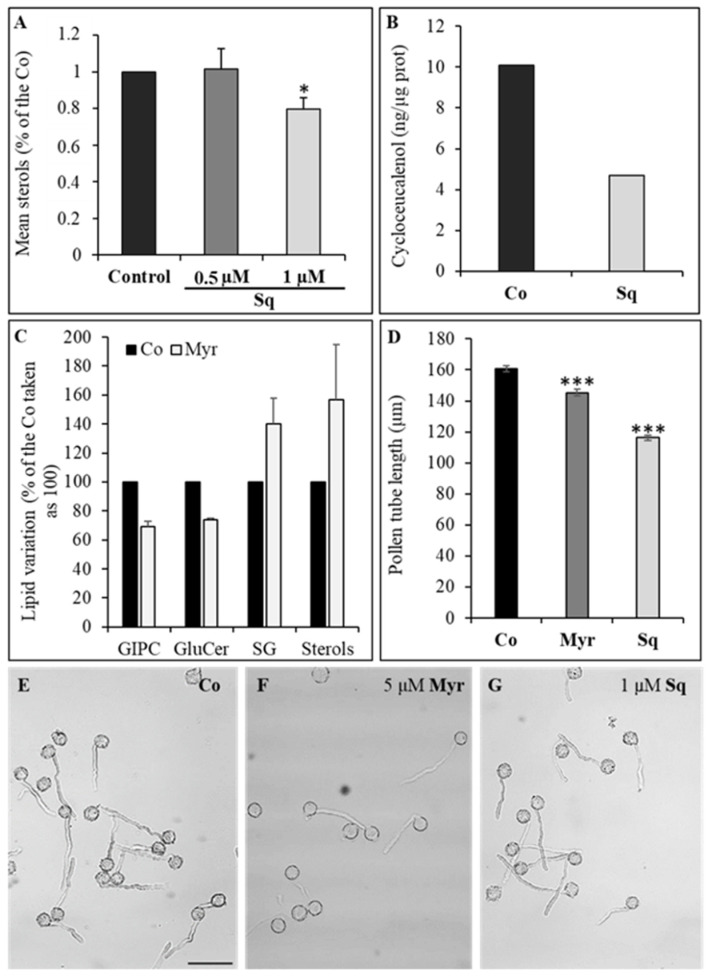
Lipid analysis and effect of sterols and sphingolipid depletion on pollen tube length. Pollen tubes were incubated with 0.5 and 1 μM squalestatin for 2 h (**A**). Only 1 μM squalestatin significantly reduced the content of sterols in the microsomal fraction (mean values as a percentage of control, n = 8; ANOVA and Tukey’s post hoc test: *p* < 0.05, reported as *). De novo synthesis of cycloeucalenol was also affected by 1 μM squalestatin ((**B**); n = 2). 5 μM myriocin significantly affected the level of Glycosy lnositol Phospho Ceramides and Glucosyl Ceramide (**C**). Furthermore, an increase in both steryl-glucoside and free sterols was observed ((**C**); mean values as a percentage of Control, n > 3). Measure of pollen tube length performed using ImageJ program (**D**) showed a decrease in pollen tube length both in presence of squalestatin and myriocin (ANOVA and Tukey’s post hoc test: *p* < 0.0001, reported as ***). Statistical analysis was performed with ANOVA. The morphology of pollen tubes does not reveal a loss of polarity at the tip in both treatments compared to Control (**E**–**G**). Scale Bar = 100 μm.

**Figure 2 plants-12-00008-f002:**
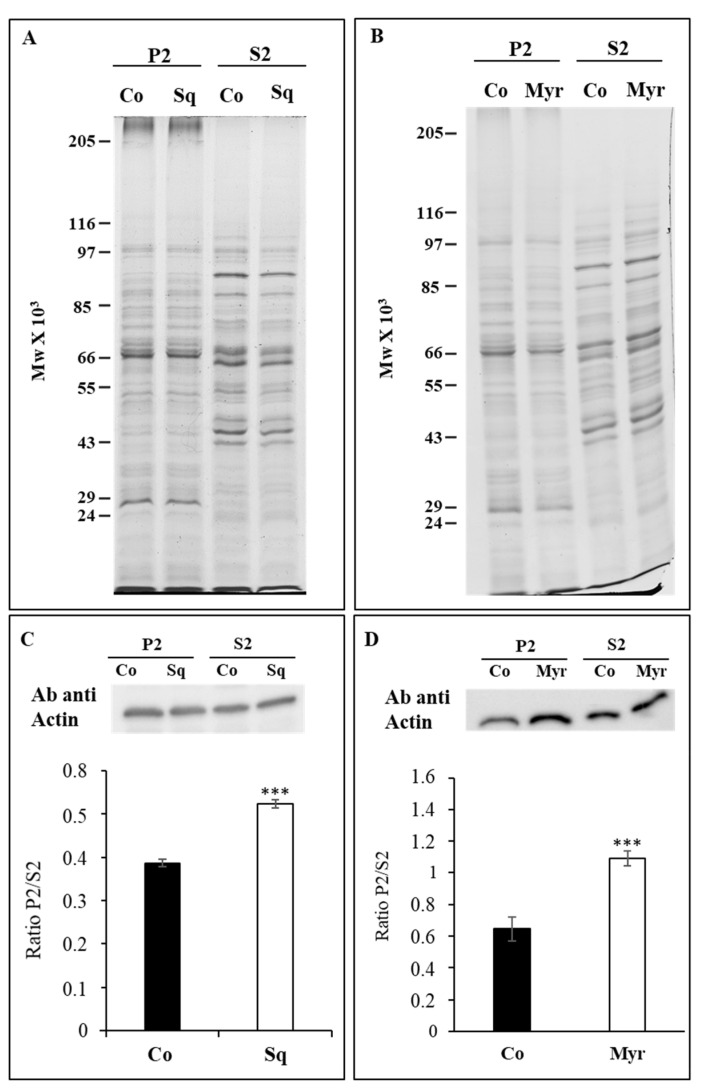
Effect of squalestatin and myriocin on actin partitioning between membranes (P2) and soluble fractions (S2). SDS-PAGE electrophoresis did not show significant differences in protein pattern between control and myriocin/squalestatin pollen tubes (**A**,**B**). However, Western blotting analysis, with an anti-actin monoclonal antibody, shows that squalestatin and myriocin significantly enhanced the interaction of actin with microsome ((**C**,**D**); western and graph; ANOVA and Tukey’s post hoc test: *p* < 0.001 reported as ***). For quantification of protein level, the area of bands was considered in 5 independent experiments and statistical analysis was performed by ANOVA.

**Figure 3 plants-12-00008-f003:**
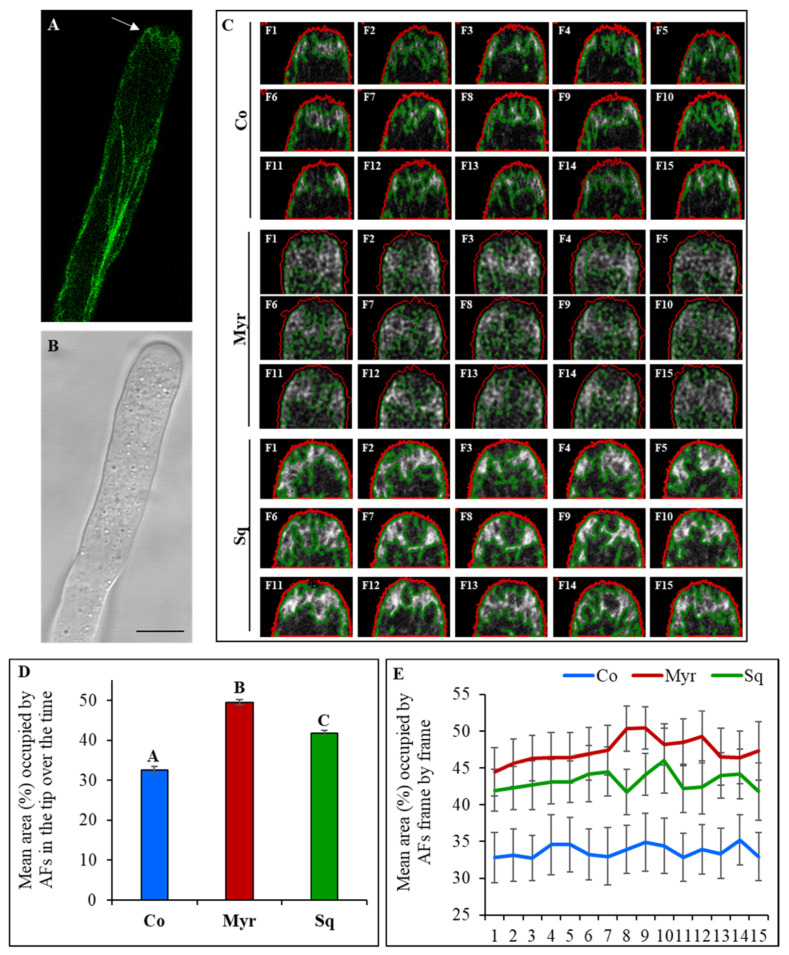
Effect of myriocin and squalestatin on actin fringe dynamic. Pollen grains were transiently transformed with a plasmid which codifies for the fusion protein Lifeact-EGFP under the control of the pollen specific promoter LAT52. In control pollen tubes in the shank and distal regions, actin longitudinal bundles were observed ((**A**) pLAT52-Lifeact-EGFP, (**B**) bright field). In the apical flanks (2–5 μm from the apical PM), short, dynamic AFs organized into a fringe ((**A**), see arrow). Since AFs longitudinal bundles were not affected by myriocin and squalestatin, analyses of cortical actin fringe in control, myriocin and squalestatin-treated pollen tubes were performed (**C**). Images were deconvolved with Huygens professional version 19.04 software and processed with CellProfiler software. ROI in red defines first 5 μm of pollen tube and area occupied by AFs is defined by the green line (**C**). The percent area (%) occupied by AFs in each tip is calculated and the mean value represented for each treatment showing higher percentage in myriocin and squalestatin-treated cells, with respect to control tubes ((**D**); ANOVA and Tukey’s post hoc test: *p* < 0.001 for Myr and *p* = 0.0055 for Sq; n > 11). Comparisons of the mean area occupied by AFs over time show that this difference is significant for each frame (**E**). Scale Bar = 10 μm.

**Figure 4 plants-12-00008-f004:**
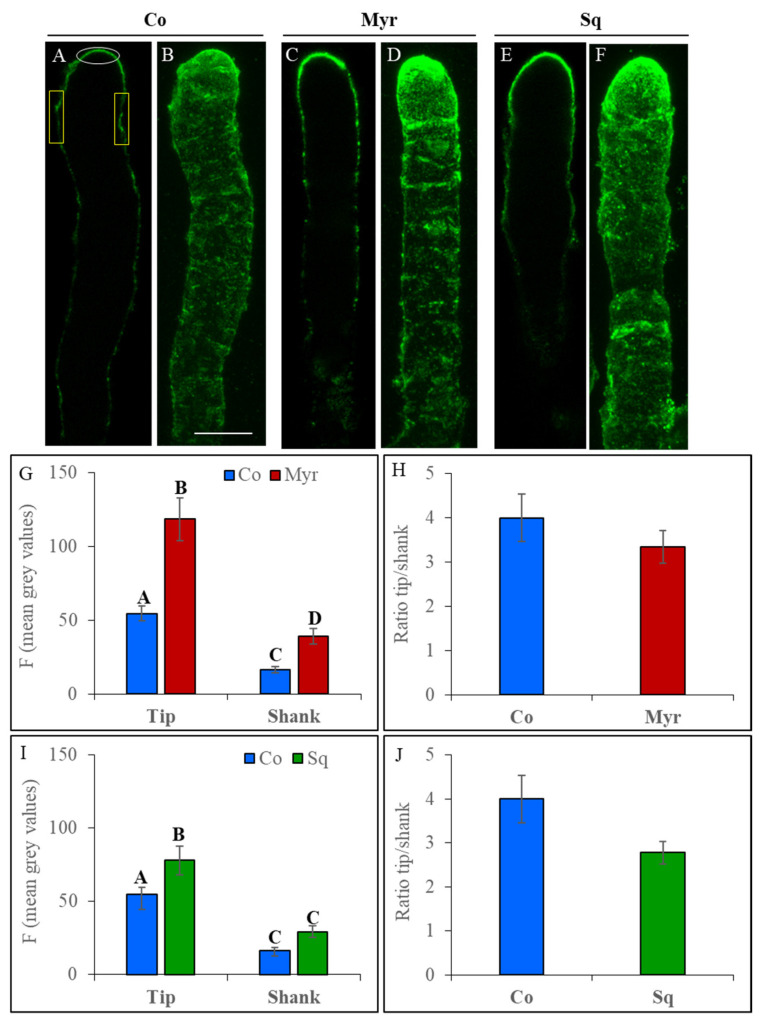
Effect of myriocin and squalestatin on pectin secretion. Distribution of methyl-esterified pectins was determined by labelling with JIM7 antibody in control (**A**,**B**), myriocin- (**C**,**D**) and squalestatin- (**E**,**F**) treated pollen tubes. Medial plane (**A**,**C**,**E**) and whole reconstruction (**B**,**D**,**F**) are shown. Quantification of fluorescence intensity in tip and shank (white and yellow ROIs, respectively, as specified in panel (**A**); n > 11) performed using ImageJ showed that myriocin and squalestatin induced an increase in esterified pectins in these regions ((**G**,**I**); tip ANOVA and Tukey’s post hoc test: *p* < 0.0001 for Myr and *p* < 0.01 for Sq; shank ANOVA and Tukey’s post hoc test: *p* < 0.05 for Myr and *p* > 0.05 for Sq). The difference in esterified pectins between tip and shank is maintained in pollen tubes grown in the presence of inhibitors (**H**,**J**). Scale Bar = 10 μm.

**Figure 5 plants-12-00008-f005:**
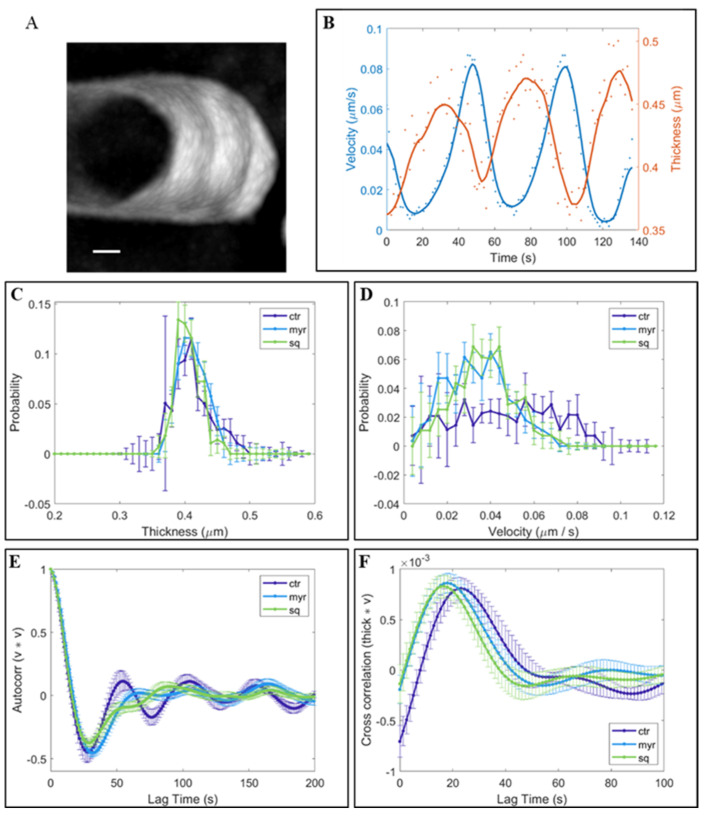
Analyses of correlation between cell wall thickness and pollen tube growth after lipid perturbation. Max projection of time lapse images of pollen tube growth (**A**). The frames where the growth is slower and the wall is thicker contributed to the higher intensity regions. Tube velocities and thicknesses in panel (**A**) are shown in the graph (**B**). Solid lines are moving averages of the raw data. The overall thickness and velocity distributions for each condition is obtained by averaging the distributions calculated on single tubes ((**C**,**D**), respectively. n > 11). Velocity/velocity autocorrelation and thickness/velocity cross correlation were obtained by averaging the correlations calculated on single tubes ((**E**,**F**), respectively). In the thickness/velocity cross correlation, the max positions of peaks are reported: Co: 23.13 ± 0.15 s, Myr: 18.65 ± 0.12 s, Sq: 17.14 ± 0.12 s. Scale Bar = 1 µm.

**Figure 6 plants-12-00008-f006:**
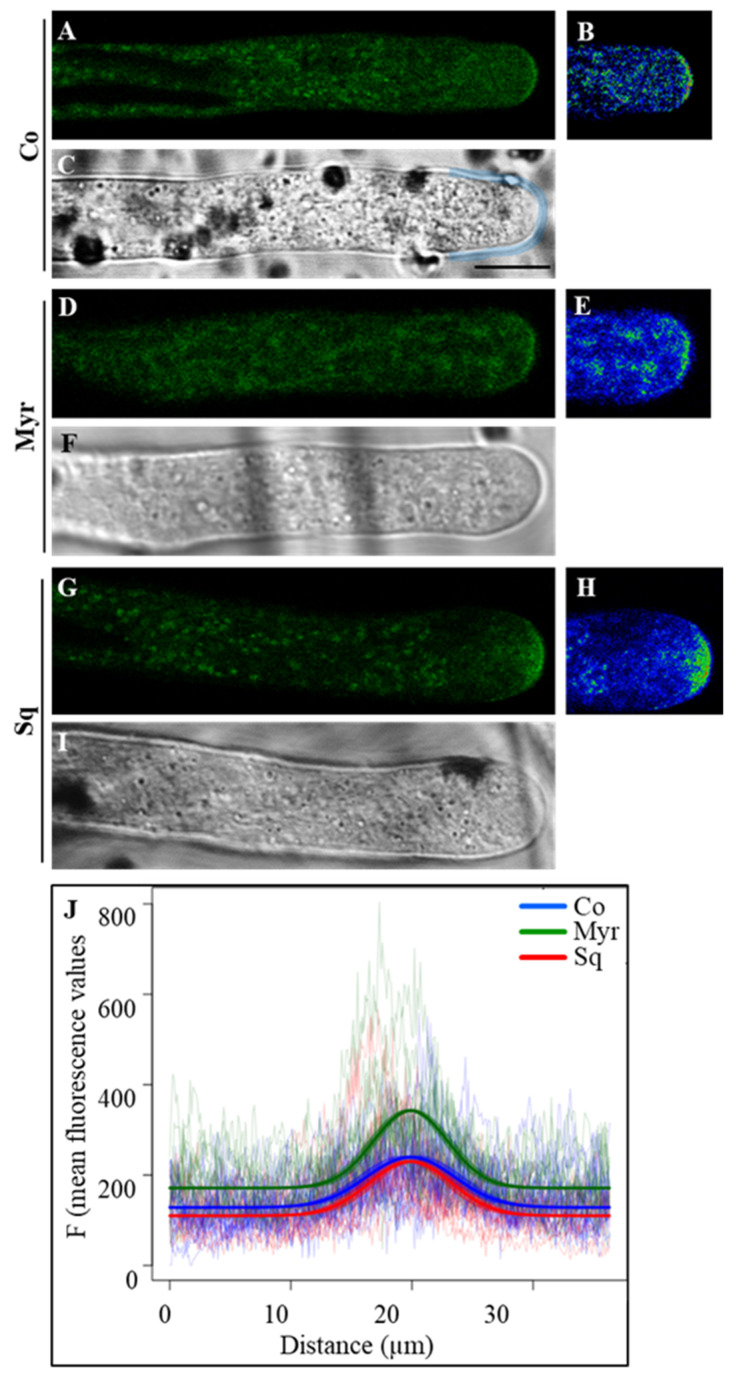
Effect of myriocin and squalestatin on NtPPME1 secretion. Pollen grains were transiently transformed using a plasmid which codifies for the fusion protein LAT52- NtPPME1-EGFP. In control and treated pollen tubes, NtPPME1 localized in the apical dome and in unidentified organelles inside the cell ((**A**,**D**,**G**) LAT52- NtPPME1-EGFP; (**B**,**E**,**H**) pseudo-colored images; (**C**,**D**,**G**) bright field for control myriocin and squalestatin pollen tubes, respectively). In squalestatin-treated pollen tubes the accumulation of NtPPME1 positive vesicles immediately behind the PM was observed (**G**,**H**). Quantitative analysis performed by ImageJ was expressed as plot profile in the apex of pollen tubes (**J**); starting about 20 µm from the tip using a freehand line as in (**C**) blue line; n > 18) and confirmed that NtPPME1 localized in the apical dome. Myriocin peak appeared narrower than control profile ((**J**); compare blue and red lines, *p* < 0.01) while squalestatin treatment induces a sharp rise of the peak in the apical dome ((**J**); compare blue and green lines; *p* < 0.01). Plot profile analyses revealed that in the shank, mean fluorescence was significantly lower in myriocin and significantly higher in squalestatin-treated pollen tubes compared to control ((**J**); ANOVA and Tukey’s post hoc test: *p* < 0.001). Bright field images (**F**,**I**). Scale bars = 10 μm.

**Figure 7 plants-12-00008-f007:**
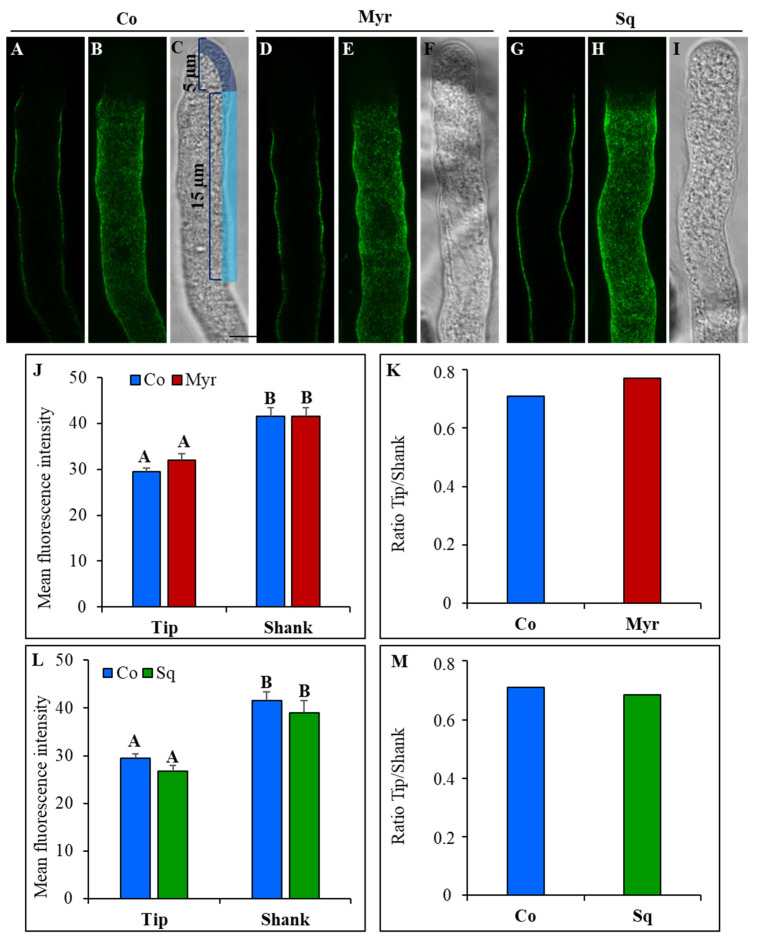
Effect of myriocin and squalestatin on cellulose deposition. Distribution of crystalline cellulose was determined by staining with GFP-CBM3A in control and treated pollen tubes ((**A**,**D**,**G**) medial plane; (**B**,**E**,**H**) whole reconstruction, (**C**,**F**,**I**) bright field for control, myriocin and squalestatin, respectively). Fluorescence intensity was measured using 20-point freehand lines in the tip and shank wall (5 µm and 15 µm, respectively, as reported in (**C**); n > 24) and plotted using ImageJ. No differences in fluorescence intensity were observed in myriocin and squalestatin-treated pollen tubes ((**J**,**L**); ANOVA and Tukey’s post hoc test: *p* > 0.05). Both myriocin and squalestatin do not modify tip/shank fluorescence ratio with respect to control cells ((**K**,**M**); ANOVA and Tukey’s post hoc test: *p* > 0.05). Scale bar = 10 μm.

**Figure 8 plants-12-00008-f008:**
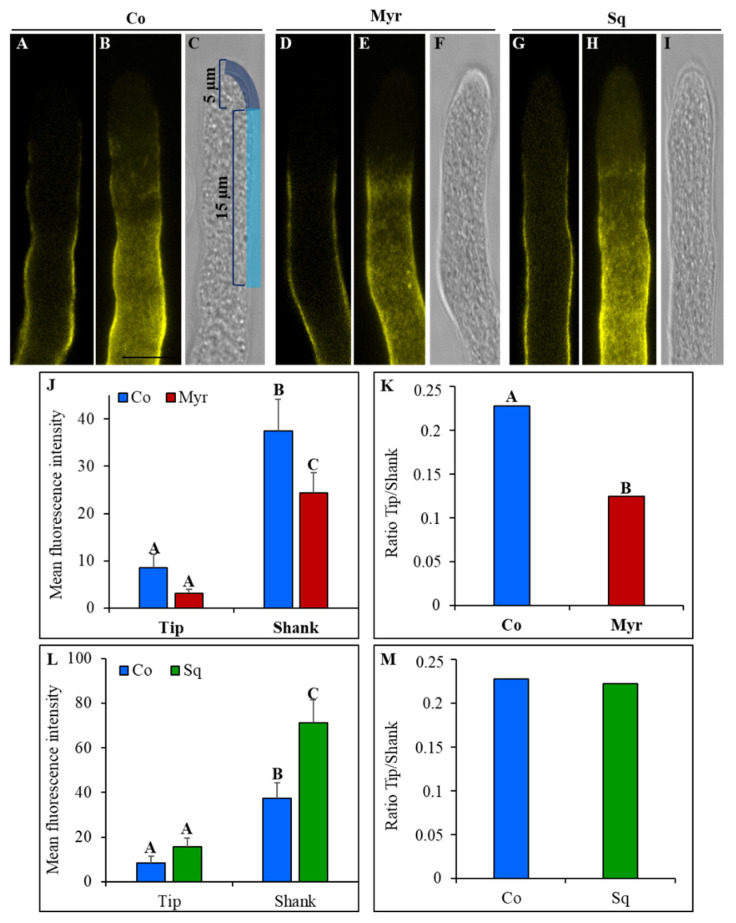
Effect of myriocin and squalestatin on callose deposition. Distribution of callose was determined by staining with aniline blue in control and treated pollen tubes ((**A**,**D**,**G**) medial plane; (**B**,**E**,**H**) whole reconstruction, (**C**,**F**,I) bright field for control, myriocin and squalestatin, respectively). Fluorescence intensity was measured using 20-point freehand lines in the tip and shank wall (5 µm and 15 µm, respectively, as reported in (**C**); n > 17) and plotted using ImageJ. The callose deposition at the tip is significantly lower in pollen tubes grown both with myriocin and squalestatin with respect to control tubes ((**J**,**L**); ANOVA and Tukey’s post hoc test: *p* < 0.005). In the shank, an opposite effect was observed since myrocin induced a decrease in callose fluorescence, while in squalestatin-treated pollen tubes fluorescence significantly increased ((**J**,**L**), respectively; ANOVA and Tukey’s post hoc test: *p* < 0.01). The tip/shank fluorescence ratio was significantly lower only in the presence of myriocin with respect to control ((**K**,**M**); ANOVA and Tukey’s post hoc test: *p* < 0.001). Scale bars = 10 μm.

**Figure 9 plants-12-00008-f009:**
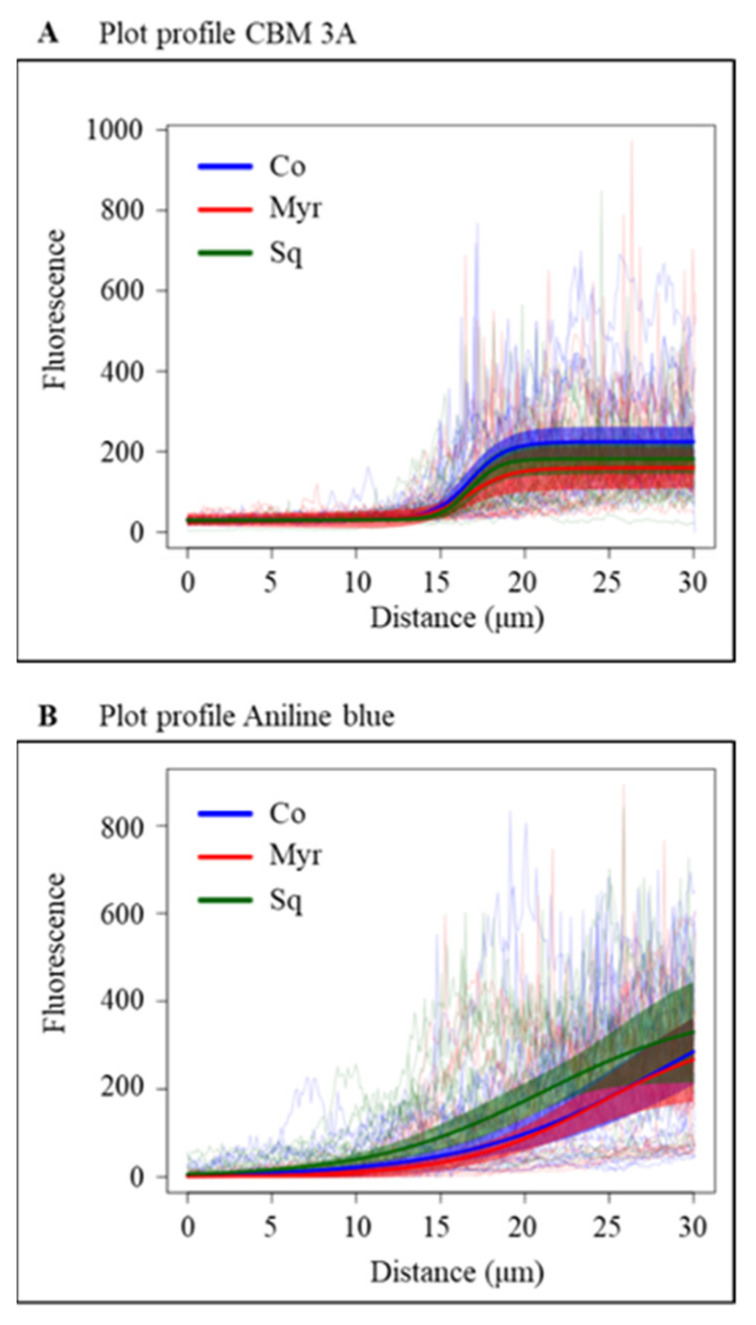
Analysis of GFP-CBM3A and Aniline Blue fluorescence distribution along pollen tubes. Each line represents the mean of grey values measured in previously analyzed pollen tubes (n > 10). The *y*-axis represents the mean fluorescence intensity, while the *x*-axis represents the distance from the pollen tube tip (0–30 μm). To measure the mean fluorescence intensity along the pollen tube edge (first 30 μm), a 20 points freehand line was plotted as in Figure 7 and Figure 8 and analyzed in more detail using ImageJ-Plot Profile. Using R statistical program, a four parameter logistic function was fitted to fluorescence data with a non-linear mixed model. Cellulose deposition was mostly affected in the distal area of pollen tubes after myriocin and squalestatin treatment compared to control ((**A**); *p* < 0.01). On the contrary, callose deposition appeared more affected in the shank area of pollen tubes. In myriocin-treated pollen tubes, the curve was flatter than that of the control (**B**). A significantly lower value of the scale parameter of the logistic curve fitted to the myriocin than control group (*p* < 0.01) indicated an initially flatter curve followed by a steeper increase. In contrast, in squalestatin-treated pollen tubes the fluorescence intensity grew more quickly toward the distal area of pollen tubes and the inflection point of the fitted logistic curve was shifted towards the tip in the squalestatin compared to the control ((**B**); *p* < 0.001).

**Figure 10 plants-12-00008-f010:**
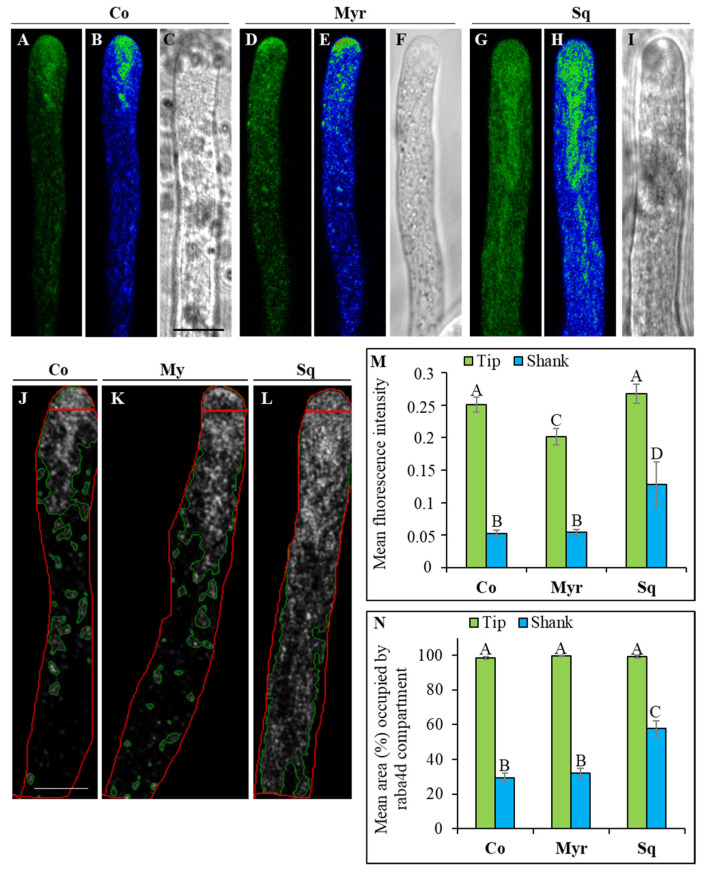
Analysis of morphology and dynamics of the clear zone. Pollen tubes were transiently transformed with LAT52-GFPRabA4d. In control cells, RabA4d localized on vesicles accumulating in the clear zone with the characteristic V-shaped morphology ((**A**) LAT52-GFPRabA4d; (**B**), pseudo-colors, (**C**) bright field). In myriocin-treated pollen tubes vesicles appear mostly concentrated in the tip dome ((**D**) LAT52-GFPRabA4d; (**E**), pseudo-colors, (**F**) bright field). With squalestatin an expansion of the clear zone with the V-shape extending more deeply from the tip to shank/distal regions is observed ((**G**) LAT52-GFPRabA4d; (**H**) pseudo-colors, (**I**) bright field). For quantitative analyses, ROIs in red define either the first 5 μm of pollen tube tip or the following 60 μm along shank and distal regions (**J**–**L**). In each ROI, the % of area occupied by RabA4d (defined by the green line) and the mean intensity for each condition were quantitatively analyzed ((**J**–**L**) for control, myriocin and squalestatin, respectively; n > 14). Myriocin induces a significant reduction in mean fluorescence intensity in the tip ((**M**); ANOVA and Tukey’s post hoc test: *p* < 0.001) and a significant increase in the mean area occupied by RabA4d compartments ((**N**); ANOVA and Tukey’s post hoc test: *p* < 0.01) with respect to control tubes. Squalestatin induced, in shank and distal regions, a significant increase in mean fluorescence intensity ((**M**); ANOVA and Tukey’s post hoc test: *p* < 0.01) and of the area occupied by RabA4d compartments ((**N**); ANOVA and Tukey’s post hoc test: *p* < 0.01). Scale bars = 10 μm.

## Data Availability

The results of this study are available on request from the corresponding author (Alessandra Moscatelli).

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
