# Peer review of "Sterols and Sphingolipids as New Players in Cell Wall Building and Apical Growth of Nicotiana tabacum L. Pollen Tubes"

_plants, 2022, doi:10.3390/plants12010008_

Round 1

Reviewer 1 Report

In this manuscript, Stroppa et al presented an interesting account on how a sterol biosynthesis inhibitor Squalestatin (Sq) and a sphingolipid biosynthesis inhibitor Myriocin (Myr) affect the lipid compositions in tobacco pollen tubes. Through a series of careful image analyses, the authors proposed that sterols and sphingolipids could serve as a link between actin filaments and polarized secretion at the growing tip of pollen tubes. I basically think the data presented are novel and of significance to the field but there are few major concerns I have for this manuscript:

(1) It falls short of showing any concrete evidence that Sq and Myr treatment has indeed changed the lipid compositions in the pollen tube membrane. Although in Figure 1, the authors showed that total sterols were reduced in 1uM Sq-treated and GIPC as well as GluCer were reduced in 5uM Myr-treated pollen tubes, these by no means are indicative of the lipid environment on the membrane of growing tip. As the authors have pointed out themselves, the plants are very much capable to activating a compensatory mechanism when sterols/sphingolipids biosynthesis is disrupted (line #174-175). Hence, changes in total sterols composition is an indirect observation. As far as I know various staining protocols have used membrane dyes such as FM4-64, LRB-PE, DiD, Bodipy-labelled C12 Sphingomyelin to visualize different lipid phases in the membrane. I strongly suggest that authors provide staining patterns using one (or combination) of these dyes in Sq-treated and Myr-treated pollen tube, ideally coupled with respective quantification to demonstrate that Sq and Myr indeed affect the sterols composition on the pollen tube membrane.

(2) Writing style is far too speculative. For example, upon observing a greater labelling of JIM7 in the Sq-treated and Myr-treated pollen tube growing tip, the authors suggested that “modification of the lipid profile could promote post-Golgi SV targeting of/fusion with the tip PM..”. Firstly, I am not aware that secretion of esterified pectins is a selective sorting event as no specific adaptors for esterified pectin-containing vesicles have been found; rather I imagine it is through bulk flow secretion as it is sensitive to Brefeldin A treatment (PMID: 25037212). Hence fusion of post-trans-Golgi network SV with the tip PM can not explain the increased and selective deposition of pectins on the arc of Sq-treated and Myr-treated pollen tube. Secondly, even if pectin secretion is through sorting, I have difficulties imagining how changes on lipid compositions on the plasma membrane could affect pectins desposition on the cell walls? Lastly, the increased HEPs deposition in Sq-treated and Myr-treated pollen tube is contradictory to the seemingly lack of changes in PME secretion of the Myr-treated pollen tubes but increased secretion/broader distribution of PME in Sq-treated pollen tubes. If PME secretion was indeed enhanced in the Sq-treated pollen tubes, wouldn’t one expect to see more LEP instead of HEP? On a related note, which PME did the authors used? Is it a tobacco or an Arabidopsis PME?  

(3) Discussion is unnecessarily lengthy and speculative. I must admit I did not enjoy reading the discussion as it feels like the whole part was lifted off from a thesis and planted there. Specifically:

Line #572-573: “the disorganization of membrane lipids by inhibition..”- there was no evidence to demonstrate the “disorganization of membrane lipids”. Please refer to point #1

Line #671: “Sterol and sphingolipid perturbation induced an increase in pectin secretion”- enhanced signal in the HEP can not be interpreted as increased pectin secretion. One could easily question why LEP signal was not enhanced if pectin secretion is generally increased?

Line #795-#886: The authors are using references from different plant species and tissue type to argue for their case. If this is a review article, I would welcome inclusion of references from various plants but doing so in the discussion part of a research article is very misleading. For example, the authors used references from Arabidopsis (ref. 85, 79, 127), Aspen (ref. 56, 128) and Leek (ref. 127) to argue for a potential role of sterols-dependent vesicular trafficking; or in an even less desirable case they cited a paper of tobacco epidermal cells which do not undergo tip growth, or an intermediate form of tip growth (ref. 101). It is important to note that what was observed in a different plant species/tissue type may not be necessary applicable in tobacco pollen tubes.

Line #818-819: “RabA4d is specifically expressed in vesicles accumulating in the clear zone” -I am not sure a gene can be expressed in vesicles?? Isnt gene expression a temporal and spatial (at tissue/cellular level) thing?

(4) Images acquired in this study was largely based on transient expression of constructs such as LAT52:PME-EGFP, LAT52: Lifeact-EGFP and LAT52:GFP-RabA4d. What approach did the author use to control for a similar expression level between samples so that conclusion drawn is not an artefact from overexpression/no expression?

Additional comments:

Figure 1.

Figure 1E: It is not clear what the control treatment is. Since Sq and Myr are both dissolved in ethanol and methanol respectively (as stated in the materials and methods) and was diluted at different factors. Please state if control is simply not-treated and if so, an appropriate control is necessary.

Figure 1E: Would be helpful to readers if Myr and Sq concentrations are stated in the figure.

Figure 1B: Lack of error bars and statistical test. Possible to provide additional biological replicate?

Figure 2A,B: What is Mrx103 ?

Figure 2C,D: Please provide Coomassie staining for the respective immunoblots to demonstrate equal loading between samples. Also, what was the statistical test used?

Figure 7: CBM3A-GFP should be GFP-CBM3A as the supplier states in their catalogue. It would be helpful if the authors could state clearly in the text what CBM3A stands for.

Minor:

(a)  Throughout the manuscript, the authors used the term “Magnification bar”. I believe it should be Scale bar?

(b)  Line22: Recent studies “have”….

Author Response

Plants

December 2th, 2022

Resubmission of research article entitled:STEROLS AND SPHINGOLIPIDS AS NEW PLAYERS IN CELL WALL BUILDING AND APICAL GROWTH OF NICOTIANA TABACUM L. POLLEN TUBES”; (Manuscript ID: plants-2010239).

by: Nadia Stroppa, Elisabetta Onelli, Patrick Moreau, Lilly Maneta-Peyret, Valeria Berno, Eugenia Cammarota, Roberto Ambrosini, Marco Caccianiga, Monica Scali and Alessandra Moscatelli

Dear Sir,

I enclose the above manuscript after revision in response to the referees' comments. We thank the reviewers for their helpful suggestions which we trust have improved the manuscript and presentation of the results and discussion. Our replies to the various points follow.

(1) It falls short of showing any concrete evidence that Sq and Myr treatment has indeed changed the lipid compositions in the pollen tube membrane. Although in Figure 1, the authors showed that total sterols were reduced in 1uM Sq-treated and GIPC as well as GluCer were reduced in 5uM Myr-treated pollen tubes, these by no means are indicative of the lipid environment on the membrane of growing tip. As the authors have pointed out themselves, the plants are very much capable to activating a compensatory mechanism when sterols/sphingolipids biosynthesis is disrupted (line #174-175). Hence, changes in total sterols composition is an indirect observation. As far as I know various staining protocols have used membrane dyes such as FM4-64, LRB-PE, DiD, Bodipy-labelled C12 Sphingomyelin to visualize different lipid phases in the membrane. I strongly suggest that authors provide staining patterns using one (or combination) of these dyes in Sq-treated and Myr-treated pollen tube, ideally coupled with respective quantification to demonstrate that Sq and Myr indeed affect the sterols composition on the pollen tube membrane.

We know that some of these dyes detect lipids in membranes. In particular, FM4-64, LRB-PE and DiD are used for whole membrane staining so they do not allow different classes of lipids to be identified.

The referee suggests Bodipy-labelled C12 Sphingomyelin for sphingolipid staining. However, Sphingomyelin is not a plant lipid and its insertion into plant membranes could increase the number/nature of liquid ordered domains as suggested in Blachutzik et al., 2012 and Yagi et al., 2021 (Blachutzik JO, Demir F, Kreuzer I, Hedrich R, Harms GS. Methods of staining and visualization of sphingolipid enriched and non-enriched plasma membrane regions of Arabidopsis thaliana with fluorescent dyes and lipid analogues. Plant Methods. 2012 Aug 6;8(1):28. doi: 10.1186/1746-4811-8-28 and Yagi N, Yoshinari A, Iwatate RJ, Isoda R, Frommer WB, Nakamura M. Advances in Synthetic Fluorescent Probe Labeling for Live-Cell Imaging in Plants. Plant Cell Physiol. 2021 Nov 10;62(8):1259-1268. doi: 10.1093/pcp/pcab104).

Moreover, use of this dye requires cell wall digestion to avoid “accumulations of the dyes within the microfibril textures of cell walls and from unspecific leaf autofluorescence signals” (Blachutzik et al., 2012). In pollen tubes, protoplasts cannot be obtained without destroying cell polarization and this impairs the study of apical growth.

However, “to demonstrate that Sq and Myr indeed affect the sterol composition on the pollen tube membrane”, we performed filipin staining and we analyzed pollen tube grown with or without Myr and Sq.

The observation of filipin was done by conventional fluorescence microscopy since confocal microscopes equipped with UV laser were not available in our facilities (all microscopes had a 405 nm laser).

The results sustained the sterol variations observed by biochemical analysis (Fig S1, p. 5, lines 173-178).

More experiments on the effect of these inhibitors on DIM structure and lipid membrane composition will be reported in a future paper, for which work is currently underway.

(2) Writing style is far too speculative. For example, upon observing a greater labelling of JIM7 in the Sq-treated and Myr-treated pollen tube growing tip, the authors suggested that “modification of the lipid profile could promote post-Golgi SV targeting of/fusion with the tip PM..”. Firstly, I am not aware that secretion of esterified pectins is a selective sorting event as no specific adaptors for esterified pectin-containing vesicles have been found; rather I imagine it is through bulk flow secretion as it is sensitive to Brefeldin A treatment (PMID: 25037212).

The delivery of methyl-esterified pectins to the apical pollen tube cell wall relies on exocytic vesicles (Chebli et al., 2012). However, the post-Golgi route does not exclude the possibility of bulk flow secretion.

Brefeldin A treatment induces a compartment formed with the contribution of Golgi endosomes and TGN, so post-Golgi trafficking is impaired. It has also been reported that alteration of degradation pathways via PVCs of endocytosed material affects pectin secretion, modifyiing the Arabidopsis pollen tube cell wall and suggesting a close link between secretion and endocytosis (Saiying Hou, Jiao Shi, Lihong Hao, Zhijuan Wang, Yalan Liao, Hongya Gu, Juan Dong, Thomas Dresselhaus, Sheng Zhong, Li-Jia Qu, VPS18-regulated vesicle trafficking controls the secretion of pectin and its modifying enzyme during pollen tube growth in Arabidopsis, The Plant Cell, Volume 33, Issue 9, September 2021, Pages 3042–3056, https://doi.org/10.1093/plcell/koab164).

The process of cell wall formation in pollen tube is therefore very complex. To avoid misunderstanding about pectin secretion processes we removed “post-Golgi” from the sentence.

Hence fusion of post-trans-Golgi network SV with the tip PM can not explain the increased and selective deposition of pectins on the arc of Sq-treated and Myr-treated pollen tube. Secondly, even if pectin secretion is through sorting, I have difficulties imagining how changes on lipid compositions on the plasma membrane could affect pectins desposition on the cell walls

We do not agree, since Sq and Myr not only altered the PM but assembled enriched sterols and sphingolipids into membranes in the TGN, so pectin secretion may be affected by lipid perturbation before they reaches the PM. The idea that lipid perturbation also affects endosomes is consistent with our observation that myr and sq treatment affected the area and trafficking of these organelles in pollen tubes. This data will be reported in our next paper.

Here, to highlight modification of secretion due to lipid perturbation, we used RabA4d, which is known to mediate trafficking of exocytic vesicles (Lee et al., 2008; Szumlanski and Nielsen, 2009; Zhou et al., 2020).

Ultimately, secretion of pectins could also be affected by actin alterations in the tip. As you can see, the situation is very complex. Further experiments into how sterols and sphingolipids affect pollen tube growth are underway.

Lastly, the increased HEPs deposition in Sq-treated and Myr-treated pollen tube is contradictory to the seemingly lack of changes in PME secretion of the Myr-treated pollen tubes but increased secretion/broader distribution of PME in Sq-treated pollen tubes. If PME secretion was indeed enhanced in the Sq-treated pollen tubes, wouldn’t one expect to see more LEP instead of HEP

We too expected these results. However, we postulate that: “As clathrin-dependent endocytosis (CDE) is involved selectively in the internalization of demethylated pectin [96-99], the decrease in LEPs in the shank of tobacco pollen tubes after lipid perturbation could be due to enhanced internalization of these pectins, allowing cell wall thickness to remain as in control pollen tubes.” (p. 22, lines 706-710).

This hypothesis is also supported by the fact that there is no increase in wall thickness, suggesting that excess cell wall was internalized, probably in the shank. Interestingly, research currently underway shows that both PVC and tubular vacuoles are affected by myr and sq treatment, in line with the close correlation between secretion and endocytosis (Hou et al., 2021, see above).

On a related note, which PME did the authors used? Is it a tobacco or an Arabidopsis PME? 

The PME used was a tobacco PME1 with a regulatory pro-region (Bosch, M.; Cheung, A.Y.; Hepler, P.K. Pectin methylesterase, a regulator of pollen tube growth. Plant Physiol. 2005 138, 1334-1346. doi:10.1104/pp.105.059865).  We replaced PME with NtPPME1 throughout the text.

(3) Discussion is unnecessarily lengthy and speculative. I must admit I did not enjoy reading the discussion as it feels like the whole part was lifted off from a thesis and planted there. Specifically:

Line #572-573: “the disorganization of membrane lipids by inhibition..”- there was no evidence to demonstrate the “disorganization of membrane lipids”. Please refer to point #1

We replaced “disorganization” with “perturbation” (p. 20, line 595). The perturbation of membrane lipids is demonstrated in Fig. 1 and S1.

Line #671: “Sterol and sphingolipid perturbation induced an increase in pectin secretion”- enhanced signal in the HEP can not be interpreted as increased pectin secretion.

The increase in methyl esterified pectins in the cell wall could be due to an increase in secretion or alteration of pectin de-esterification by PME.

However, 1) alteration of pectin de-esterification was excluded by maintenance of the ratio of HEP fluoresce in the tip/shank. Secreted HEPs were normally de-esterified. 2) PME seemed to lose its pro-region earlier in treated pollen tubes than in control cells, explaining why more HEPs were de-esterified. 

Modification of pectin secretion therefore seems the most probable hypothesis, sustained by alteration of the actin fringe and Rab4D distribution/dynamics. 

One could easily question why LEP signal was not enhanced if pectin secretion is generally increased?

See above

Line #795-#886: The authors are using references from different plant species and tissue type to argue for their case. If this is a review article, I would welcome inclusion of references from various plants but doing so in the discussion part of a research article is very misleading. For example, the authors used references from Arabidopsis (ref. 85, 79, 127), Aspen (ref. 56, 128) and Leek (ref. 127) to argue for a potential role of sterols-dependent vesicular trafficking; or in an even less desirable case they cited a paper of tobacco epidermal cells which do not undergo tip growth, or an intermediate form of tip growth (ref. 101). It is important to note that what was observed in a different plant species/tissue type may not be necessary applicable in tobacco pollen tubes.

Unfortunately, very few studies have been performed in pollen tubes, so we are obliged to refer to previous studies in somatic cells.

Regarding the “even less desirable case”, if a review has been published in a journal, it is just as appropriate for citation as any other paper.

Line #818-819: “RabA4d is specifically expressed in vesicles accumulating in the clear zone” -I am not sure a gene can be expressed in vesicles?? Isnt gene expression a temporal and spatial (at tissue/cellular level) thing?

We changed “RabA4d is specifically expressed” to “RabA4d is specifically localized…” (p. 24 line 812-813).

The Discussion and Conclusions have been edited to reduce their length and eliminate speculative passages.

(4) Images acquired in this study was largely based on transient expression of constructs such as LAT52:PME-EGFP, LAT52: Lifeact-EGFP and LAT52:GFP-RabA4d. What approach did the author use to control for a similar expression level between samples so that conclusion drawn is not an artefact from overexpression/no expression?

In pollen tubes where proteins were not expressed, undetectable fluorescence was observed by microscopy.

First, all analyses of treated samples were compared to the control sample under the acquisition parameters reported in Material and Methods: “Nor were differences in control and treated pollen tubes due to different protein expression, since pollen tubes with similar fluorescent intensity were considered during acquisition. (p. 28, lines 1004-1006)… To compare different experimental conditions, live data mode acquisitions were always performed with the same laser intensity and PMT settings”. (p. 29, lines 1018-1020). Only actively growing pollen tubes were considered.

Overexpression was only observed with LAT52:Lifeact-EGFP, see supplementary figure 2. Fluorescence in these pollen tubes was very high for the acquisition parameters.

Expression of LAT52:NtPPME1-EGFP was only obtained at high plasmid concentrations as suggested by Bosh et al. (2005). Overexpression was never observed, as reported in Results: “To detect fluorescence, we used high plasmid concentrations [72]. NtPPME1 localization in pollen tubes matched that described in the literature [72] and changes in fluorescence pattern were attributed to the drug treatments. (p. 14, lines: 406-408).

LAT52:GFP-RabA4d was used at plasmid concentrations already published in Idilli et al. (2013) in tobacco pollen tubes. This is the lowest concentration necessary to observe RabA4D localization, as reported in Results: “In control pollen tubes, the pattern of RabA4d distribution appeared as already reported [76, 77]. At the plasmid concentration used, RabA4d did not overexpress and changes in fluorescence pattern were attributed to the drug treatments.” (p. 18, lines 534-536).

Additional comments:

Figure 1.

Figure 1E: It is not clear what the control treatment is. Since Sq and Myr are both dissolved in ethanol and methanol respectively (as stated in the materials and methods) and was diluted at different factors. Please state if control is simply not-treated and if so, an appropriate control is necessary.

Stock solutions of sq and myr are concentrated; we used very few µl of methanol or ethanol. We performed control experiments in the analysis of pollen tube length and no variation was observed, probably due to evaporation of the two components in the pollen culture. We added a sentence in Results: “The same amount of ethanol (Sq solvent) and methanol (Myr solvent) was added to cul-ture media during pollen tube growth in order to analyze if they affected pollen tube growth. No alteration of pollen tube length was observed (data not shown).” (p. 5, lines: 196-199)

Figure 1E: Would be helpful to readers if Myr and Sq concentrations are stated in the figure.

We included the concentrations in Figure 1 F, G.

Figure 1B: Lack of error bars and statistical test. Possible to provide additional biological replicate?

The data reported in Fig 1B concerns two analyses. Unfortunately, a third analysis is not possible now because the equipment on the facility is no longer available.

Figure 2A,B: What is Mrx103 ?

Mr indicates markers of molecular weight so we replaced Mr with Mw (Fig. 2).

Figure 2C,D: Please provide Coomassie staining for the respective immunoblots to demonstrate equal loading between samples. Also, what was the statistical test used?

The corresponding amount of protein loaded on the blotted gel is shown in panels A and B of the same figure and is based on the Bradford protein assay.

Figure 7: CBM3A-GFP should be GFP-CBM3A as the supplier states in their catalogue. It would be helpful if the authors could state clearly in the text what CBM3A stands for.

We replaced CBM3A-GFP with GFP-CBM3A throughout the text.

Minor:

(a)  Throughout the manuscript, the authors used the term “Magnification bar”. I believe it should be Scale bar?

We replaced Magnification with Scale throughout the text.

(b)  Line22: Recent studies “have”….

We corrected the sentence.

The paper has already been revised by native English-speaking translator of scientific texts.

We trust that the paper is now in order for publication in Plants.

Yours faithfully,

Alessandra Moscatelli

Reviewer 2 Report

This manuscript describe a comprehensive series of experiments designed to test the hypothesis that lipid rafts may link actin dynamics and polarized secretion in pollen tube growth. This is an interesting topic and certainly of interest to readers of “plants”.

The authors have used a combination of lipid biosynthesis inhibitors, lipid biochemistry, transient transformation with molecular markers, monoclonal antibodies and advanced imaging techniques to pose and answer a number of questions.

They initially identify dosage levels of the inhibitors that perturb the lipid composition in pollen tubes but do not disrupt growth polarity (though they do affect growth in terms of tube length).

They then go on to assess the effect of these inhibitors on actin dynamics, cell wall deposition (independently assessing pectin, cellulose and callose), and the dynamics and morphology of the “clear zone” in the cytoplasm of pollen at the tube tips. I find the set of experiments provided to be comprehensive, informative and carefully performed. In my opinion, the results presented provide good support for the authors conclusions that sterols and sphingolipids may play a significant role in role in modulating actin filament organization and dynamics in the pollen tube tip, and also in mediating the pulsing growth phases exhibited by pollen tubes. Overall this is an excellent piece of work!

The manuscript has been carefully prepared – but I do have a few specific comments that should be addressed prior to publication (most are grammatical or minor typographical errors).

Specific comments:

Line 5 “Scali5and”

- add space

Line 25  “organisms evolutionary distant“

- grammar “organisms evolutionarily distant“

Line 32  “thus indicating a“

- “thus suggesting a”, indicating seems a little strong......

Line 47 Arabidopsis thaliana,

-       full Latin name please italicize

Line 49 recycling and endocytosis, emerging as important

-       grammar add “are” to read “are emerging”

Line 55 Coherently, use of the styryl

-       “Coherently” isn’t really used in this way, I suggest “In agreement” or “Consistent with this”.

Line 92-95 Pectin de-esterification with cellulose and callose deposition prevent expansion of the cell wall into the shank and distal region in response to turgor pressure, thus maintaining the cylindrical shape of pollen tubes, especially in the difficult journey through the style [53].

-       As written this states that the maintenance of the pollen tube cylinder happens especially in the style. Surely this occurs all the time, but is more challenging in the : difficult journey through the style”?

Line 108 PM, preferring detergent resistant “

-       The use of the word preferring suggests that CalS has a choice in where it localizes to. A netter phrase would be “preferentially localizing to detergent resistant....” (sorry I know it is the same word used in a different way – English is a weird language!)

Line 168 with respect to control (Fig. 168

-       With respect to controls (or the control)

Line 233 used, overexpressed pollen tubes were occasionally observed

-       Pollen tubes cannot be over-expressed. They over-express a gene or construct. Here I suggest “pollen tubes over-expressing Lifeact-EGFP”

Line 239 In LIFEACT-EGFP overexpressed pollen tube

-       In LIFEACT-EGFP overexpressing pollen tubes”

Line 388 Since PME1 was never overexpressed, changes in fluorescence pattern were assigned to the drug treatments.

– no basis is provided for the judgement that PME1 was never over-expressed and it depends rather on one’s definition of over-expressed. As it was being ectopically expressed as well as endogenously it could be argued that it is definitely being over-expressed. However I assume this statement is based on a lack of phenotype? Please explain....

Line 732 The localization of PME in control tobacco pollen tubes was coherent with localization of the enzyme carrying the pro-region

– suggestion “consistent” rather than “coherent”?

Line 801 such as Golgi and TGN.

– add the - “such as Golgi and the TGN.”

Line 933 MgSO4, 0.5 mM CaCl2, 3.7% formaldehyde

– missing subscripts, MgSO4 and CaCl2

Line 956 5.4Lipid analysis

– missing space

Line 1080 The fists ones were used to.....

– I am not 100% certain what this should be, should it read “first ones”?

Line 1102 L.M

– period missing after the M

Line 1103 V.B and E.C.

– period missing after the B

Line 1104 M.S performed

– period missing after the S

Line 1110 Galles, UK

– should be “Wales, UK”

Line 1112 Prof. PK. Hepler (Massachusetts University, Amherst, USA)

- period missing after P, P.K. and the institution should be University of Massachusetts, Amherst, USA.

Line 1128 232-245

- period missing at the end

Line 1145 f-actin

– capitalize the F

Line 1272 Plants 2013 2, 211-229.

- 2013 should be in bold

Line 1314 Gerbeau-Pissot, P.;, Lherminier, J.;

– delete comma after P.;

Line 1322 f-actin

– capitalize the F

Line 1412 Henis, Y.I, Lewinsohn, E.;

– period missing after Y.I

Line 1440 Plant Cell 2008 20, 3312-3330.

- 20 should be in italics

Author Response

Plants

December 2th, 2022

Resubmission of research article entitled:STEROLS AND SPHINGOLIPIDS AS NEW PLAYERS IN CELL WALL BUILDING AND APICAL GROWTH OF NICOTIANA TABACUM L. POLLEN TUBES”; (Manuscript ID: plants-2010239).

by: Nadia Stroppa, Elisabetta Onelli, Patrick Moreau, Lilly Maneta-Peyret, Valeria Berno, Eugenia Cammarota, Roberto Ambrosini, Marco Caccianiga, Monica Scali and Alessandra Moscatelli

Dear Sir,

I enclose the above manuscript after revision in response to the referees' comments. We thank the reviewers for their helpful suggestions which we trust have improvedthe manuscript and presentation of the results and discussion. Our replies to the various points follow.

All the suggestions have been accepted.

– no basis is provided for the judgement that PME1 was never over-expressed and it depends rather on one’s definition of over-expressed. As it was being ectopically expressed as well as endogenously it could be argued that it is definitely being over-expressed. However I assume this statement is based on a lack of phenotype? Please explain....

We agree with the reviewer's interpretation of “overexpression”. By “overexpression” we mean a mislocalization of PME due to an excess of protein expression. Expression of LAT52:NtPPME1-EGFP was only obtained with high amount of plasmid as suggested by Bosh et al., 2005. Mislocalization of PME1 was never observed in pollen tubes, as we report in Results: “NtPPME1 localization in pollen tubes matches that already described in the literature [72] and changes in fluorescence pattern were attributed to the drug treatments." (p. 14, lines: 406-408). Only actively growing pollen tubes were considered.

We trust that the paper is now in order for publication in Plants.

Yours faithfully,

Alessandra Moscatelli

Reviewer 3 Report

Dear author and the editor,

I think this an interesting work, tiacylglycerol is continuously synthesized during pollen tube growth, sphingolipids and sterols present a very unique composition in pollen. In my opinion, although there spend much more time for this study, there are still some problems to be answered and improved.

I. Please consier the title name, rigorously.

II. More description about background pollen tubes in abstract, which should done in introduction or discussion. And as well the same problem happend in conclusion, there are too burdensome description, so please summarized this part and provide highlights of this study.

III. Please simplify the text in results, in which just show the data and profiles fo your funding objectively rather than with more discussion statments. 

III. In my opinion, all the data in the figures and tables should be statistically analyzed based on one-way anova or variance test, please unified this performance in the whole text.

Author Response

Plants

December 2th, 2022

Resubmission of research article entitled:STEROLS AND SPHINGOLIPIDS AS NEW PLAYERS IN CELL WALL BUILDING AND APICAL GROWTH OF NICOTIANA TABACUM L. POLLEN TUBES”; (Manuscript ID: plants-2010239).

by: Nadia Stroppa, Elisabetta Onelli, Patrick Moreau, Lilly Maneta-Peyret, Valeria Berno, Eugenia Cammarota, Roberto Ambrosini, Marco Caccianiga, Monica Scali and Alessandra Moscatelli

Dear Sir,

I enclose the above manuscript after revision in response to the referees' comments. We thank the reviewers for their helpful suggestions which we trust have improvedthe manuscript and presentation of the results and discussion. Our replies to the various points follow.

  1. Please consier the title name, rigorously.

Sorry, but we don’t understand the reviewer's request.

  1. More description about background pollen tubes in abstract, which should done in introduction or discussion. And as well the same problem happend in conclusion, there are too burdensome (onerose, abbondanti) description, so please summarized this part and provide highlights of this study.

We edited part of the pollen tube description in the abstract.

III. Please simplify the text in results, in which just show the data and profiles fo your funding objectively rather than with more discussion statments.

We removed sentences that could be considered discussion from Results.

III. In my opinion, all the data in the figures and tables should be statistically analyzed based on one-way anova or variance test, please unified this performance in the whole text.

Statistical analysis has now been performed by Anova. We added the new statistical results to the manuscript, specifying that Anova was the method used.

We trust that the paper is now in order for publication in Plants.

Yours faithfully,

Alessandra Moscatelli

Round 2

Reviewer 1 Report

The authors have answered all my questions.

Just one last thing- please change CDE (clathrin-dependent endocytosis, lines #960, #964) to the more commonly used CME (clathrin-mediated enocytosis, lines #970, #972). I suppose the authors meant the same thing?

Author Response

Resubmission of research article entitled:STEROLS AND SPHINGOLIPIDS AS NEW PLAYERS IN CELL WALL BUILDING AND APICAL GROWTH OF NICOTIANA TABACUM L. POLLEN TUBES”; (Manuscript ID: plants-2010239).

by: Nadia Stroppa, Elisabetta Onelli, Patrick Moreau, Lilly Maneta-Peyret, Valeria Berno, Eugenia Cammarota, Roberto Ambrosini, Marco Caccianiga, Monica Scali and Alessandra Moscatelli

Dear Sir,

We thank you for the helpful suggestions.

About your comment:

Just one last thing- please change CDE (clathrin-dependent endocytosis, lines #960, #964) to the more commonly used CME (clathrin-mediated enocytosis, lines #970, #972). I suppose the authors meant the same thing?

Yes, we meant the same thing. We replace CDE with CME (P. 22, line 706, 707 and 710).

We trust that the paper is now in order for publication in Plants.

Yours faithfully,

Alessandra Moscatelli

Reviewer 3 Report

Most problems in the manuscript have been revised and I suggest accept this study in present form.

Author Response

Resubmission of research article entitled:STEROLS AND SPHINGOLIPIDS AS NEW PLAYERS IN CELL WALL BUILDING AND APICAL GROWTH OF NICOTIANA TABACUM L. POLLEN TUBES”; (Manuscript ID: plants-2010239).

by: Nadia Stroppa, Elisabetta Onelli, Patrick Moreau, Lilly Maneta-Peyret, Valeria Berno, Eugenia Cammarota, Roberto Ambrosini, Marco Caccianiga, Monica Scali and Alessandra Moscatelli

Dear Sir,

We thank you for the helpful suggestions which we trust have improved the manuscript and presentation of the results and discussion.  

Yours faithfully,

Alessandra Moscatelli
